


# A multi-disciplinary analysis of the exceptional flood event of July 2021 in central Europe. Part 1: Event description and analysis

Susanna Mohr[1,2], Uwe Ehret[1,3], Michael Kunz[1,2], Patrick Ludwig[1,2], Alberto Caldas-Alvarez[2], James E. Daniell[1,4], Florian Ehmele[2], Hendrik Feldmann[2], Mário J. Franca[3], Christian Gattke[5], Marie Hundhausen[2], Peter Knippertz[2], Katharina Küpfer[1,2], Bernhard Mühr[1], Joaquim G. Pinto[1,2], Julian Quinting[2], Andreas M. Schäfer[1,6], Marc Scheibel[7], Frank Seidel[3], and Christina Wisotzky[1,8]

[1]Center for Disaster Management and Risk Reduction Technology (CEDIM), Karlsruhe Institute of Technology (KIT), Karlsruhe, Germany
[2]Institute of Meteorology and Climate Research (IMK-TRO), Karlsruhe Institute of Technology (KIT), Karlsruhe, Germany
[3]Institute for Water and River Basin Management, Karlsruhe Institute of Technology (KIT), Karlsruhe, Germany
[4]Institute of Photogrammetry and Remote Sensing, Karlsruhe Institute of Technology (KIT), Karlsruhe, Germany
[5]Erftverband, Bergheim, Germany
[6]Geophysical Institute, Karlsruhe Institute of Technology (KIT), Karlsruhe, Germany
[7]Wupperverband, Wuppertal, Germany
[8]Institute of Economics, Karlsruhe Institute of Technology (KIT), Karlsruhe, Germany

**Correspondence:** Susanna Mohr (mohr@kit.edu)

**Abstract.** The July 2021 flood in central Europe was one of the five costliest natural disasters in Europe in the last half century with estimated total damage of EUR 32 billion. This study investigates the complex interactions between meteorological, hydrological, and hydro-morphological processes and mechanisms that led to the exceptional flood. Furthermore, we present our estimates of the impacts in terms of inundation areas, traffic disruptions, and economic losses. The estimation of inundation

5   areas as well as the derived damage assessments were carried out during or directly after the flood, and show the potential of near-real-time forensic disaster analyses for crisis management, emergency personnel on-site, and the provision of relief supplies. The superposition of several factors resulted in widespread extreme precipitation totals and water levels well beyond a 100-year event: slow propagation of the low pressure system *Bernd*, convection embedded in a mesoscale precipitation field, unusually moist air masses associated with a significant positive anomaly in sea surface temperature over the Baltic Sea, wet

10  soils, and steep terrain. Various hydro-morphodynamic processes as well as changes in valley morphology observed during the event exacerbated the impact of the flood. Relevant effects included, among many others, the occurrence of extreme landscape erosion, rapidly evolving erosion and scour processes in the channel network and urban space, recruitment of debris from the natural and urban landscape, deposition and clogging of bottlenecks in the channel network with eventual collapse. This study is part one of a two-paper series. The second part puts the July 2021 flood into a historical context and into the context of

15  climate change.



# 1 Introduction

The severe flood event in western Germany and neighboring regions in mid-July 2021 was one of the most severe natural catastrophes in Europe in the last half century. Precipitation totals of up to 150 mm over an extensive area falling within 15 to 18 hours led to the severe flooding on 14 and 15 July 2021. It claimed many lives with at least 180 deaths in Germany alone. The two German federal states of North Rhine-Westphalia (NRW) and Rhineland-Palatinate (RP) were particularly affected, but also the neighboring countries of Belgium, Netherlands, and Luxembourg. In Germany, the flood had a devastating effect in the north-east of the low mountain range Eifel, where villages along the rivers Ahr and Erft, both left tributaries of the Rhine, were affected (Fig. 1). Severe damage was caused to buildings, household goods, industry, but also to infrastructure such as railways, roads, and bridges. According to Munich Re (2022), total losses amount to EUR 46 billion, EUR 33 billion of which in Germany alone, making this event the most expensive natural disaster in Germany to date. The insured portion was relatively low at EUR 8.2 billion in Germany (all countries EUR 11 billion; Munich Re, 2022) because of a low insurance density of residential buildings (only about 37 to 47 % are insured) and infrastructure damage.

The Center for Disaster Management and Risk Reduction Technology (CEDIM, www.cedim.kit.edu, last access: 9 May 2022), an interdisciplinary research center in the field of disasters, risks, and security at Karlsruhe Institute of Technology (KIT), Germany, has been conducting Forensic Disaster Analyses (FDA) in near-real time since 2011 (e.g., Kunz et al., 2013; Merz et al., 2014; Piper et al., 2016; Wilhelm et al., 2021). The approach of forensically investigating disasters stems from the interdisciplinary research program Integrated Research on Disaster Risk (IRDR) and their program Forensic Investigation of Disasters (FORIN; Burton, 2010). IRDR uses the forensic approach to uncover the root causes of disasters through in-depth investigations and to build an understanding of how natural hazards do or do not become disasters. CEDIM aims for a timely analysis within a few hours to days. CEDIM's FDA examines the dynamics and interrelations of disasters, identifies major risk drivers, estimates the impact (damage, fatalities, displaced), and infers possible implications for disaster mitigation– as was the case in July 2021 on the extraordinary flood event, for which the report with first damage estimates was issued only 1 week after the event (Schäfer et al., 2021). The authors emphasized (cf. KIT, 2021) that the flood risk in the affected region was significantly underestimated, as current flood hazard maps do not yet contain any historical data, but only consider homogeneous gauge measurements of the last decades (i.e., considered period < 50 years).

Since the release of the FDA report, several other studies on the July 2021 flood have been published: besides first meteorological analyses by the weather services in the affected countries (Junghänel et al., 2021; Dewals et al., 2021; MeteoLux, 2021), the World Weather Attribution (WWA) initiative, a consortium of international scientists to identify possible impacts of climate change on current extreme weather events, has already produced a report in August 2021 (Kreienkamp et al., 2021). The authors of the report show that climate change has increased the likelihood of a precipitation event with meteorological characteristics similar to the July 2021 event that affected the rivers Ahr, Erft, and Meuse by 1.2 to 9 times, whereas the intensity of the maximum 24 h rainfall has increased by about 3 to 19 %. In addition, they estimated a return period of 400 years for the rainfall event under present climatic conditions. Faranda et al. (2022) also performed an attribution study of the flood event, using an approach based on atmospheric circulation analogs that allows atmospheric dynamics to be taken into account. They





found that the persistence index they defined increased over the recent period, suggesting that recent cut-off lows in Western Europe tend to remain more stationary, allowing for longer lasting precipitation events with an increased risk of flooding. Because some of the gauging stations were partially or completely destroyed during the July 2021 flood, the actual discharges were initially unclear. Specifically for the river Ahr, where two severe historical floods had already occurred in 1804 and 1910 (Roggenkamp and Herget, 2014a, b), Roggenkamp and Hergert (2022) reconstructed the peak discharges on the basis of histor-

ical data and background knowledge about the topography. They quantified a maximum runoff between 1000 and 1200 $\mathrm{m^3\,s^{-1}}$ and classified the event to be on a similar level as the flood of 1804, thus being very rare but not unique. Based on a field trip to the Ahr Valley, Korswagen et al. (2022) discussed some structural failures of buildings. They observed only few cases of structural failure due to hydrostatic flood pressure alone, which is usually neutralized by the water inside the flooded buildings. Rather, erosion and damming of debris, as well as scouring caused by flood water flow were identified to be relevant to struc-

tural failure. This led to the partial or total collapse of several masonry buildings, making vertical evacuation infeasible. The study of Koks et al. (2021) gives a first overview of the impact on large-scale critical infrastructure systems and the state of recovery in the first 6 months after the event. The authors show that several critical infrastructures in Germany and Belgium were severely damaged or completely destroyed–ranging from destroyed bridges and sewage systems to severely damaged schools and hospitals. In autumn 2021, Fekete and Sandholz (2021) drew first lessons from the failure of (early) warning chains and

the insufficient effectiveness of German preventive and protection measures. The authors emphasize that the floods have once again demonstrated the importance to identify communication problems in warning chains; a central problem are persistent gaps in knowledge interpretation and communication. Using data of the July 2021 flood, Thiebes and Schrott (2021) discussed in general how early warning systems work or fail, what they can achieve and where possible weaknesses lie.

To better assess, predict, and manage natural disasters, such as the July 2021 flood, close collaboration and interaction

among scientists and practitioners across discipline boundaries are absolutely vital–as demonstrated by the wide variety of disciplines involved in the aforementioned studies. A multi-disciplinary framework such as FDA enables assessing the complex interactions of processes across different compartments from meteorology (weather situation, precipitation) to hydrological conditions (river basins, flow characteristics) to hydro-morphological impacts (changes in valleys morphology, erosion and deposition areas) to impacts on assets and environment. The objective of this two-part study is a multi-disciplinary assessment

of the entire process chain of the July 2021 flood in central Europe–from causes to impacts to historical classification and climatological context. While Part 1 focuses on the description of the event across various disciplines (meteorological, hydrological, hydro-morphological, economic), the second part (Ludwig et al., in prep., henceforth referred to as PART2) puts the event in both historical and anthropogenic climate change contexts.

The paper is structured as follows: Sect. 2 introduces the different data sets and methods used. Sect. 3 discusses the prevail-

ing meteorological and hydrological particularities and characteristics before and during the July 2021 flood, followed by a discussion of the hydro-morphodynamic processes in relation to the flood. Sect. 4 addresses the impacts and consequences with a focus on near-real-time estimation of inundation areas, damage assessment, and rail and road infrastructure failure. Finally, Sect. 5 discusses and summarizes the main results, draws conclusions, and provides an outlook on PART2.


## 2  Data and methods

The region most affected in July 2021 by the heavy rainfall and subsequent flooding (henceforth referred to as study area) is a low mountain range named Eifel with steep, deeply carved valleys. It is located in western Germany (mainly in the federal state of RP), eastern Luxembourg, and south-eastern Belgium (see Fig. 1). In the west, the Eifel is limited by the river Meuse, in the south by the river Moselle, and in the east by the river Rhine. To the west, the Eifel is continued by the low mountain region of the Ardennes in Belgium and to the north by the lowland region of the Cologne Bay in Germany. In the east, the study area extends beyond the Eifel and across the Rhine. In total, the study area covers roughly $20\,000\,\text{km}^2$ ($150 \times 130\,\text{km}$), which is unusually large for the registered high precipitation intensities and totals (see Sect. 3.1) and a key characteristic of this event.

In the following, districts, towns, and municipalities frequently mentioned in the manuscript are briefly introduced: Bad Neuenahr-Ahrweiler, a town in northern RP, is the capital of the Ahrweiler district, which is crossed by the river Ahr. The Ahrweiler district is divided into several municipalities, such as Ahrbrück, Altenahr (Fig. 1), Antweiler, Dernau, Insul, Rech, and Schuld. The Euskirchen district (with its name-giving town, Fig. 1), neighboring the Ahrweiler district in the northwest, is located in the southwest of NRW. The rivers Erft, Ahr, Kyll, and Urft have their sources in the Euskirchen district. In the downstream reaches of the Erft (i.e., north of the Euskirchen district) lies the district of Rhein-Erft-Kreis with the town of Erftstadt. While the Erft and Ahr flow directly into the Rhine, the Kyll first runs into the Moselle and the Urft into the Rur, which afterwards flows into the Meuse.

### 2.1  Data

#### 2.1.1  Precipitation data

To quantify precipitation amounts during the flood event, hourly and daily precipitation totals at single ground-based observational stations were taken from the network operated by Deutscher Wetterdienst (DWD) comprising about 2000 stations in Germany. The temporal resolution of the precipitation measurements is usually 1 hour, but can reach up to 1 minute. Unless otherwise noted, daily precipitation readings cover the period from 05:50 to 05:50 UTC to match with other data sets used in this study (see below).

In addition, different gridded precipitation data sets provided by DWD were used in this study for both the event analysis and the estimation of return periods: daily HYRAS data (*Hydrometeorologische Rasterdatensätze*; Rauthe et al., 2013) and hourly RADOLAN data (*Radar-Online-Aneichung*; Weigl and Winterrath, 2009; Winterrath et al., 2018). HYRAS is a gridded data set covering Germany and its relevant river basins in neighboring countries at a $5 \times 5\,\text{km}^2$ grid resolution currently available for the period from 1951 to 2015 (update in preparation). It is based on several thousand climate stations interpolated to the regular grid considering elevation, exposition, and climatology. A sub-sample of HYRAS is the HYRAS-DE data, formerly known as REGNIE (*Regionalisierte Niederschlagshöhen*), covering only Germany but with a higher resolution of $1\,\text{km}^2$. HYRAS-DE is continuously updated on a daily basis and thus available for the July 2021 event. Caldas-Alvarez et al. (2022) found

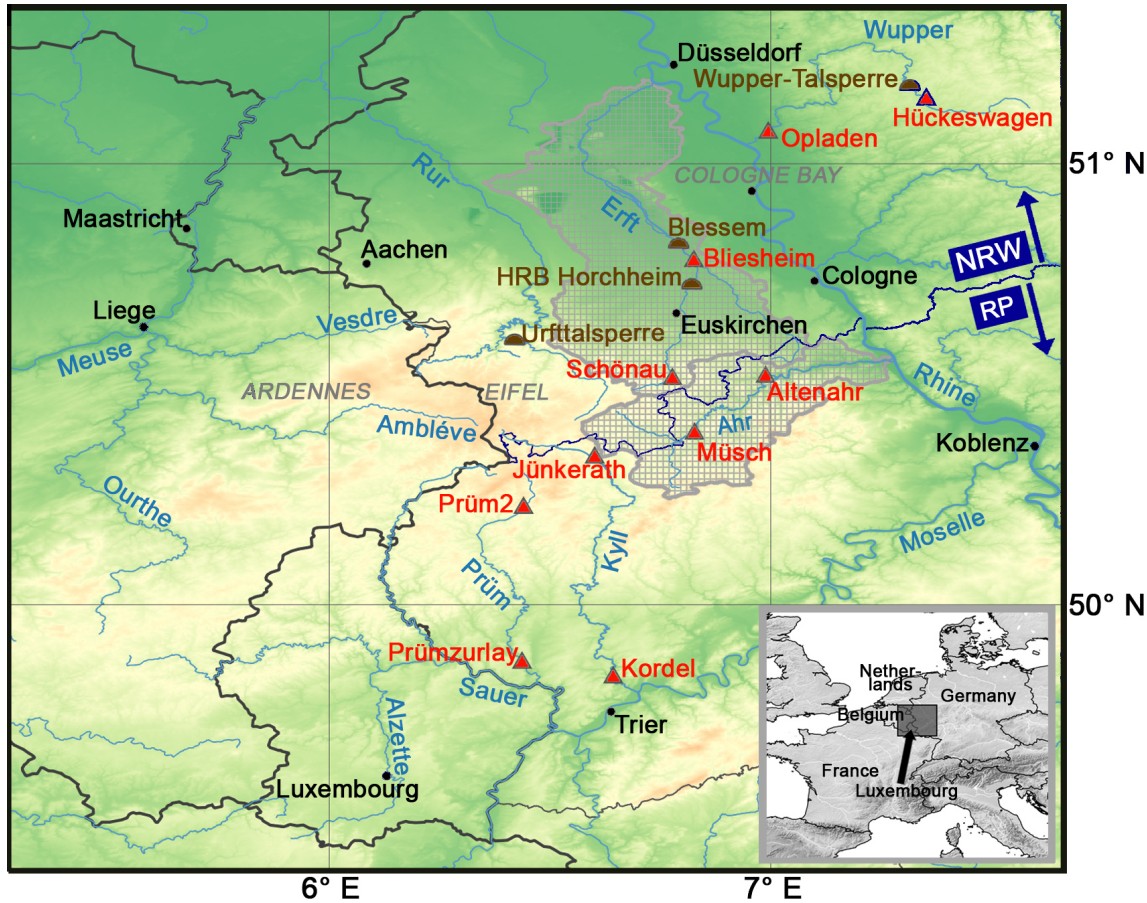

**Figure 1.** Overview map of the study area with European context (inset at small bottom right) with the main rivers (blue), river gauges (red), and reservoirs (brown) addressed in the paper, including the catchments of the rivers Ahr and Erft (hatched gray). In addition, the low mountain range Eifel, the Ardennes (both part of the Rhenish Slate Mountains), and the low terrace plain Cologne Bay (gray capital letters) and the border between the two federal states North Rhine-Westphalia (NRW) and Rhineland-Palatinate (RP) are labeled (dark blue).

that HYRAS-DE reflects well the absolute frequency of precipitation observations and is well suited for process-based and statistical analyses of extreme precipitation.

RADOLAN is a radar-based near-real-time precipitation data set covering Germany and parts of the neighboring countries with a horizontal resolution of roughly $1 \, \text{km}^2$ and an hourly temporal resolution and is available since 2001. Measured reflectivities $Z$ of the 17 radar sites operated by DWD are converted into precipitation rates $R$ using seasonal differentiated empirical $Z$-$R$ relations. To account for uncertainties in this conversion as well as for typical radar artifacts, the radar-based precipitation rates are calibrated using hourly data of over 1000 ground-based observational stations by applying different weighting and comparative techniques.





### 2.1.2 Weather forecast and analysis data

Data from the Global Environmental Multiscale (GEM) model (Côté et al., 1998) were used to analyze the geopotential pattern and precipitable water over Europe during the flood event. The GEM model, an integrated weather forecasting and data assimilation system, has been developed by the Recherche en Prévision Numérique (RPN), Meteorological Research Branch (MRB), and the Canadian Meteorological Centre (CMC). It is a global weather forecast model that is routinely use by the CMC. In its global uniform resolution configuration it is run twice daily (initialization times: 00:00 and 12:00 UTC). The

output data set has a horizontal resolution of 15 km, the forecast period is 240 hours, the forecast time step is 3 hours.

The predictability of the event is assessed by operational forecasts with the ICON model by DWD for different horizontal resolutions and lead times. The ICON model has been introduced into DWD's operational forecast system in January 2015 (Zängl et al., 2015). In this study, we used the regional refinements of the global ICON forecast over Europe (ICON-EU, 7 km horizontal grid spacing) and Germany (ICON-D2, 2.2 km grid spacing) as well as output from the ICON-D2 ensemble pre-

diction systems (ICON-D2-EPS). The considered ICON-EU simulations are initialized at 00:00, 06:00, 12:00, and 18:00 UTC and provide forecasts for the next 120 hours. The higher resolved ICON-D2 simulations are initialized every 3 hours (00:00, 03:00 UTC, etc.) and provide 27 hour forecasts. In addition, the DWD produces ensemble forecasts with the ICON-D2-EPS based on 20 ensemble members (Reinert et al., 2020), for the same initial times and domain as ICON-D2.

Additionally, the Extreme Forecast Index (EFI) for precipitation based on the ensemble prediction systems (EPS) of the

European Centre for Medium-Range Weather Forecasts (ECMWF) was considered to assess the possibility of the occurrence of extreme weather. The EFI indicates whether the ECMWF-EPS forecast distribution is substantially different from the model climate (more details about EFI see, e.g., Lalaurette, 2003; Zsoter et al., 2015). An EFI of 0.5 to 0.8 indicates an unusual event, and EFI values above 0.8 indicate a very unusual or extreme event.

### 2.1.3 River gauge data

We collected data (water levels $W$ and streamflow $Q$) of several river gauges in the study area. They were selected with the objective of (a) covering the study area–from the river Wupper in the east to the river Amblève in the west, (b) covering a range of basin sizes–from $31.9\,\mathrm{km^2}$ at gauge Schönau/Erft to $816\,\mathrm{km^2}$ at gauge Kordel/Kyll, (c) covering the position along streams–wherever possible we selected two gauges per river, one in the headwater and one close to its junction (see Fig. 1; river names in blue, gauge names in red, reservoir names in brown). Additionally, we selected gauges of special interest, such

as gauge Schönau/Erft for its proximity to retention basin Horchheim (see Sect. 3.2), gauge Bliesheim/Erft for its proximity to the pit at Blessem (urban district of the town Erftstadt; see Sect. 3.3), and gauge Altenahr/Ahr for the morphological effects in the river Ahr (see Sect. 3.3) and its historical context (cf. PART2). Water level data come from direct observations; streamflow data are mainly based on water level observations and gauge-specific water level-discharge relations ($W$-$Q$-relations). In cases where water level data were either not available (mainly due to gauge destruction), water levels exceeded the existing $W$-$Q$-

relations, or $W$-$Q$-relations became invalid because of morphological changes of the river bed, streamflow data were estimated





by hydraulic models or nearby gauges (for details see Sect. 3.2). All data were provided by the water administrations of Rhineland-Palatinate, the Erftverband and the Wupperverband. A summary of the gauge data is given in Table 1 (see Sect.3.2).

### 2.1.4 Satellite data

Data of the earth observation satellites Sentinel-1 and Sentinel-2 of the Copernicus program (Sentinel Hub, 2021) were used
to estimate the floodplain. Sentinel data are available on an almost daily basis. Depending on the current ascension, new imagery is available every second or third day. Sentinel-1 provides synthetic aperture radar imaging independent of weather conditions with a maximum spatial resolution of 5 m. These data allowed to utilize imagery as early as 15 July, independent of the occurring cloud cover. In contrast, Sentinel-2 provides true color imagery with a maximum resolution of 10 m, which is sensitive to weather conditions. Thus, initial imagery of Sentinel-2 from 16 July could not be used due to extensive cloud cover.
First clear sky imagery was available from 18 July. The Sentinel-2 imagery proved more useful due to its true color imagery, as the Sentinel-1 radar images could not resolve most of the inundation areas (see Sect. 4).

### 2.1.5 Traffic data

In order to estimate medium-term effects of the flood on transportation infrastructure, Deutsche Bahn AG (DB) provided data on rail traffic disruptions from mid-July to mid-October 2021 for NRW and from mid-July to mid-September 2021 for RP.
The DB data were pre-filtered and contain only flood-related reports. The number of disrupted lines on certain key dates in the period under consideration (approx. every 4 weeks) was evaluated.

For the same time period, road traffic data were used, which are equivalent to the reports issued by the police to, for example, radio stations. We obtained the information from the internet platform www.Stau1.de (last access: 9 May 2022), which provides historical congestion and traffic reports from Germany. As there was no pre-filtering for the road data regarding the flood, we
developed our own filter that contains flood-related words. This ensured that all traffic reports that explicitly refer to the flood are included. Note that if traffic reports did not contain flood-related words but were induced by the flood, they were not evaluated. In order to check the validity of the data, the analysis was supplemented by quality controls. Reports for the same route and route section were combined.

## 2.2 Methods

### 180 2.2.1 Trajectory analysis

The pathway of air masses that reached the affected region was investigated with 10-day kinematic backward trajectories calculated from ECMWF's ERA5 reanalysis data (Hersbach et al., 2020) using the Lagrangian Analysis Tool (LAGRANTO; Sprenger and Wernli, 2015). The trajectories are based on the three-dimensional (3D) wind field ($u$, $v$, $\omega$) on all model levels at a horizontal grid spacing of $0.5° \times 0.5°$ and temporal resolution of 3 hours. The backward trajectories were initialized every
6 hours between 14 July 06:00 UTC and 15 July 06:00 UTC at four grid points (51.0, 50.5, 50.0, 51.0 °N; 7.0, 7.0, 6.5, 6.5 °E;





see Fig. 1). Following Sodemann et al. (2008), we started the trajectories vertically every 30 hPa between 970 and 590 hPa and considered only those, which exhibit a relative humidity greater than 80 % at their initial time.

### 2.2.2 Extreme value statistics

In both parts of this two-paper series, extreme value statistics was used to estimate return periods $T_{\mathrm{RP}}(x)$ of precipitation events

$x$ (or vice versa, $x_{\mathrm{RV}}(T)$) from both observations (PART1) and model data (PART2). Making use of the Peak-over-threshold approach (POT; Wilks, 2006), only precipitation events above the 95 % percentile ($p_{95}$) were considered for the analysis. To this upper tail of the distribution, a three-parameter Generalized Pareto Distribution (GPD) was fitted given by the cumulative distribution function $F_{\mathrm{GPD}}$ (e.g., van Montfort and Witter, 1986; Coles et al., 2001):

$$F_{\mathrm{GPD}}(x) = 1 - \left(1 - \frac{k\,(x - \xi)}{\alpha}\right)^{1/k} , \qquad (1)$$

where the input variable $x$, for example, are daily precipitation totals. The scale parameter $\alpha$ and the shape parameter $k$ resulted from the fit using the maximum likelihood estimation (MLE). The statistical uncertainty of the return periods (95 % confidence interval) was estimated using a bivariate normal distribution (Kotz and Nadarajah, 2000). The location parameter $\xi$ was defined by the chosen threshold (i.e., $\xi = p_{95}$). The statistical relation between $F$ and the corresponding return period $T$ can be expressed as $T = [\lambda \cdot (1 - F)]^{-1}$ (e.g., Madsen et al., 1997; Brabson and Palutikof, 2000), where $\lambda$ is the crossing rate

(average number of events per year). Using this relation and Eq. (1), the return values for a certain return period $x_{\mathrm{RV}}(T)$ and the return period of a specific value $T_{\mathrm{RP}}(x)$ are defined by:

$$x_{\mathrm{RV}}(T) = \xi + \frac{\alpha}{k} \cdot \left[1 - (\lambda T)^{-k}\right] , \qquad (2a)$$

$$T_{\mathrm{RP}}(x) = \lambda^{-1} \cdot \left[1 - \frac{k}{\alpha} \cdot (x - \xi)\right]^{-1/k} . \qquad (2b)$$

Adjusting statistical distribution functions to a data series allows for estimating high return periods beyond the available

time period. Furthermore, the comparatively strong noise or high random component of an empirical return period estimation (e.g., block maximum) is reduced to a certain degree (e.g., Bezak et al., 2014).

### 2.2.3 Antecedent Precipitation Index

Antecedent wetness conditions in a river basin can be a decisive factor whether a rainfall event becomes a flood or not. Therefore, we used the established Antecedent Precipitation Index (API; Kohler and Linsley, 1951; Viessman et al., 2002) to

describe the conditions prior to the event. The usability of API as a soil moisture proxy was shown, for example, by Blanchard et al. (1981) or Teng et al. (1993). API is based on a weighted precipitation accumulation over several days, typically 7 to 14 days (Heggen, 2001). In line with Schröter et al. (2015), we used a precondition period of 30 days in this study. The weighting factor $k$ was set to $k = 0.9$ as suggested by Heggen (2001) or Schröter et al. (2015). The spatial and temporal resolution of API depend on the input data. In this study, API was applied to the $1\,\mathrm{km}^2$ RADOLAN data (see Sect. 2.1.1)

aggregated to a daily temporal resolution.



## 3   Event description and analysis

Several factors at various spatial and temporal scales played an important role for the catastrophic flood event in July 2021. Based on analyses of model and observational data and complemented by own simulations and assessments, we discuss in the following the complex interactions between meteorological (Sect. 3.1), hydrological (Sect. 3.2), and hydro-morphodynamic
(Sect. 3.3) processes and mechanisms.

### 3.1   Meteorological aspects

#### 3.1.1   Atmospheric conditions prior and during the flood event

**Synoptic overview and atmospheric characteristics**

At the beginning of the second decade in July 2021, a prominent upper-level trough was located over the Atlantic Ocean (not
shown). Showing cut-off tendency, the upper-level trough moved quickly southeastward and was analyzed about 300 kilometers west of Ireland on 11 July 00:00 UTC, increasingly turning into a closed mid- and upper-level low pressure system. The next day, the trough structure with its meridional oriented axis extended from the western North Sea across the English Channel and the western half of France to the Balearic Islands. As characteristic of upper-level troughs in a westerly flow, it moved eastward until it was blocked by a quasi-stationary anticyclone located over northeastern Europe. This blocking anticyclone was already
present in the region during the 3 preceding weeks (not shown). The associated cloud-free conditions and high solar insolation contributed to unusual large sea surface temperature (SST) anomalies of up to 8 K over the Baltic Sea (Fig. 2b).

Ahead of the approaching upper-level low pressure system, the air pressure began to drop and a surface low pressure system named *Bernd* entered the central European weather stage. *Bernd* and the associated frontal system moved first very slowly eastward, with its center moving from southern Switzerland/northwestern Italy across the Alps towards southern Germany, and
at later stages retrograde (westward). A low-level pressure trough with an embedded frontal system formed over the North German Plain (dashed 1010 hPa contour in Fig. 2a). It was exactly that area, in which the most intense rainfall occurred. Analyses of the middle troposphere reveal a short-wave trough on the northwestern flank of the low pressure system (solid contour in Fig. 2a), which provided an additional atmospheric uplift. The extensive central European low pressure complex included extremely moist air masses in its circulation (colored areas in Fig. 2a). Over northern Germany, very high values of
total precipitable water of more than $40 \, \mathrm{kg \, m^{-2}}$ were reached, which occur only very rarely.

These very high values of total precipitable water suggest that the moisture-laden air masses eventually contributed to the extreme precipitation totals in the affected region. In order to investigate the pathway of the moisture-laden air mass and to identify possible source regions of moisture prior to the event, we evaluated the 10-day history of air masses with the trajectory approach (see Sect. 2.2.1). As we were particularly interested in moisture uptake due to surface evaporation, we focused on
air parcels at the time when they were located in the planetary boundary layer (PBL). The density of air parcels located in the PBL over the entire 10-day period is shown in Figure 2b. The majority was located over northern central Europe, the North Sea, and the Baltic Sea. From there, the air masses were transported towards the affected region on the northern flank of *Bernd*. Considering all air parcels in and above the PBL, a similar spatial distribution was obtained (not shown). The large fraction

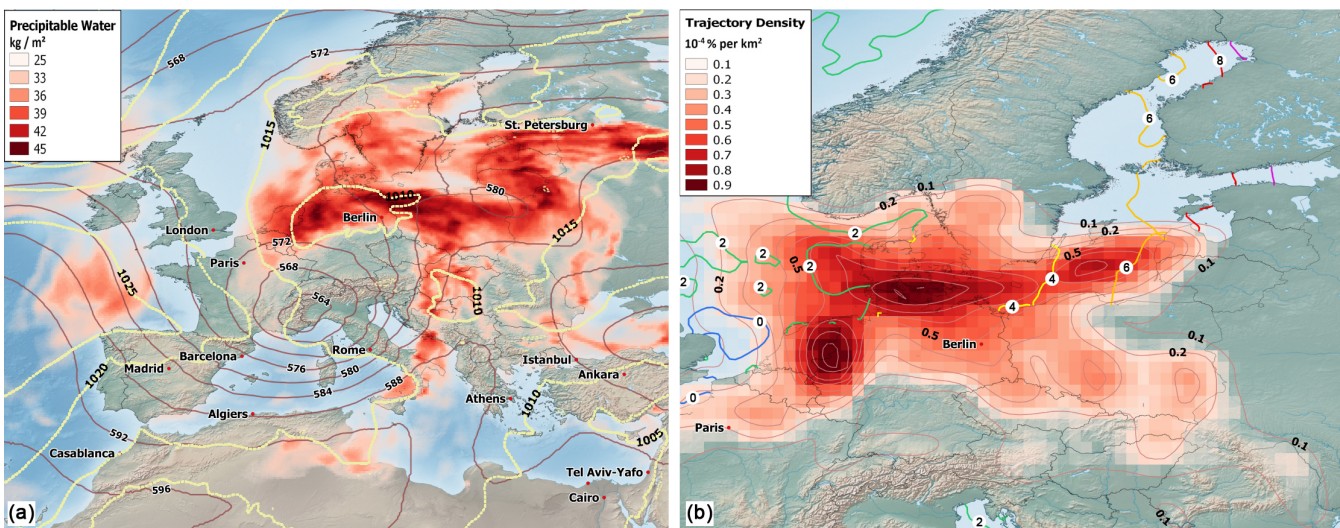

**Figure 2.** (a) 500 hPa geopotential (solid contours in gpdm), mean sea level pressure (dashed contours in hPa), and total precipitable water (shading in kg m$^{-2}$) on 14 July 2021 12:00 UTC (GEM analysis). (b) Density of air parcels (based on trajectories analysis) in the PBL over the entire 10-day period (shading and red contours every 0.1 % per area of $10^4$ km$^2$ starting at 0.1 % per area of $10^4$ km$^2$; based on ERA5); the outermost contour encloses 67 % of all air parcels. SST anomaly relative to the climatology (ERA5, 1971–2000; averaged from 8 to 15 July 2021; colored contours in K). Both backgrounds made with © Natural Earth.

over the North and Baltic Seas is noteworthy since both were characterized by unusual sea surface temperature anomalies of
1 to more than 6 K during this period (colored contours in Fig. 2b). This indicates that surface evaporation over the North Sea
and Baltic Sea served as major moisture source prior to the event. Heavy precipitation associated with quasi-stationary low
pressure systems, their fronts, or convective systems located on the western flank of persistent blocking systems is common in
Europe during summertime, so the large-scale situation is not unusual (Piper et al., 2016; Mohr et al., 2019, 2020; Kautz et al.,
2022). In the next section, we discuss the evolution of the precipitation fields in detail.


**Precipitation fields and statistics**

The most intensive precipitation areas and their displacement showed the influence of upper-level shortwave troughs already
on 13 July (see Fig. S1a in the Supplementary material). Over southwestern Germany, a large area with widespread heavy
precipitation formed in the morning of 13 July. This happened in accordance with the cyclonic rotation around the upper-level
low pressure area, and the precipitation areas first moved northeastward, then northward, before turning westward over northern
Germany and finally finding its way into NRW and RP.

The heavy precipitation event leading to the devastating floods was predominantly stratiform in nature, but with embedded
areas of convective heavier rain. The rain began during the early morning hours on 14 July, and started at the DWD weather
station in Cologne-Stammheim around 04:00 UTC (see Fig. S2 in the Supplementary material). In the Ahr catchment at the
DWD weather station of Bad Neuenahr-Ahrweiler, located near the mouth into the Rhine, rainfall started around 05:00 UTC.

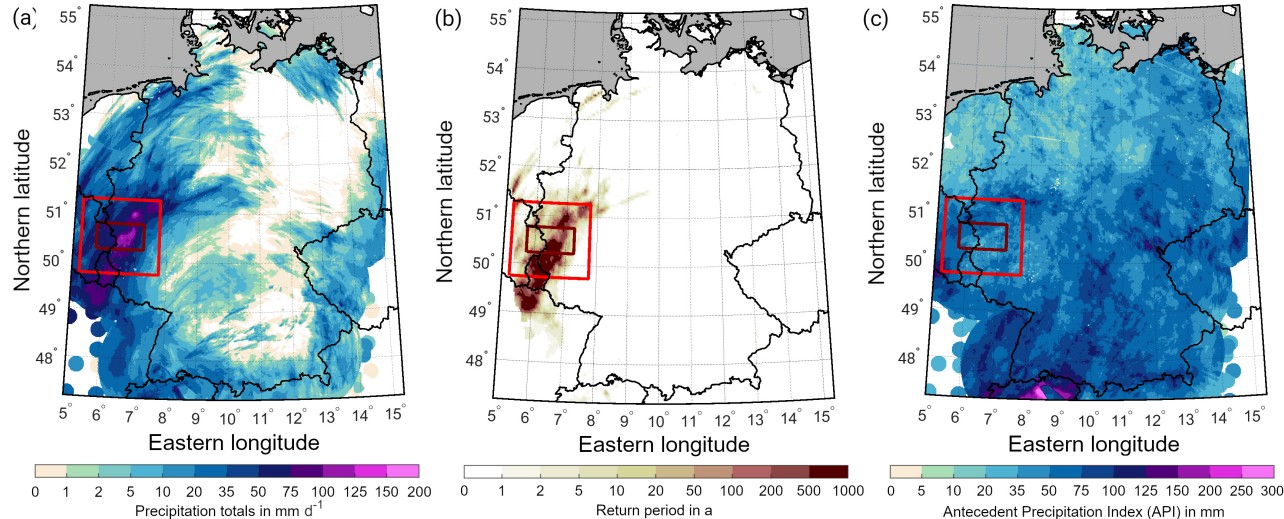

**Figure 3.** Event characteristics with (a) 24 h precipitation totals based on RADOLAN data (14 July 05:50 UTC to 15 July 2021 05:50 UTC), (b) statistical return periods of (a), and (c) the Antecedent Precipitation Index (API) estimated from RADOLAN on 14 July 05:50 UTC. Note that the RADOLAN data have been remapped to the 5 km HYRAS grid in (b) as the climatological values are taken from HYRAS (reference period: 1951 to 2015). The larger light red rectangle indicates the region named LReg, the smaller dark red rectangle the region named SReg.

Initially, it rained only lightly and irregularly, but covering almost the entire catchment area from about 09:30 UTC. Around noon, precipitation intensified, and between 13:30 and 17:30 UTC, the highest precipitation intensities were observed in the study area. From 17:30 UTC onwards, precipitation attenuated from the east, and before midnight, the rain event ceased in the affected river catchments. Thus, the major part of the precipitation totals in the study area fell within approximately 15 hours

on 14 July.

During this period, precipitation totals of up to 150 mm were recorded, for example, at Cologne-Stammheim (see Fig. S2 and Table S1 in the Supplementary material). The maximum hourly precipitation intensity reached 33 mm. The long-term average for the month of July at this station is 69 mm (1981–2010); thus, in just a few hours, the rainfall added up to more than twice the usual monthly precipitation. Further measurements at DWD stations resulted in precipitation totals between 62 mm near

the river Rhine and 144.8 mm in the Euskirchen district at the edge of the northern Eifel (station Kall-Sistig; see Table S1 in the Supplementary material). Thus, the rain amounts exceeded the usual monthly July precipitation at most of the stations.

The large spatial extent of the high precipitation area can also be clearly seen in the RADOLAN data (Fig. 3a). The study area is largely represented by the larger light red rectangle in Figure 3 (w.r.t. PART2 hereafter LReg), while the smaller dark red rectangle (hereafter SReg, cf. PART2 Sect. 2 for details) covers the highest precipitation totals and the mainly affected

Ahr catchment. On average, the 24 h precipitation totals on 14 July according to RADOLAN was 55.4 mm in the LReg and 75.2 mm in the SReg. On the same day, high precipitation totals of more than 50 mm were also observed over larger areas in the adjacent regions of Belgium and Luxembourg. Especially in these areas and also in the northern part of LReg, precipitation totals of 20 to 50 mm have already been observed the day before (see Fig. S1a in the Supplementary material) resulting in 48 h





totals (until 15 July 05:50 UTC) of above 75 mm (Fig. S1b in the Supplementary material). Consequently, the spatial average

values of the 48 h precipitation totals are significantly higher: 74.4 mm for LReg and 91.1 mm for SReg, respectively.

For a first climatological classification of the precipitation event, return periods were calculated using the GPD with the POT method (see Sect. 2.2.2) based on 65 years of HYRAS data as reference. For this purpose, the 24 h precipitation totals of RADOLAN were interpolated on the HYRAS grid. The rather large area with high precipitation totals is reflected in high return periods (Fig. 3b) exceeding over 100 years. Especially in the southern part of the study area near the border between

Germany and Luxembourg, observed rainfall totals are exceptionally high in a statistical context. On average, the return period is approximately 700 years (95 % confidence interval 40 to $2 \cdot 10^{8}$ years) in LReg and 800 years (95 % confidence interval 50 to $4 \cdot 10^{7}$ years) in SReg. This is the same order of magnitudes as in other studies. For example, Kreienkamp et al. (2021) estimated a return period at least of about 400 years for an event with similar magnitude and extent occurring anywhere in the area in western Europe between the Alps and the Netherlands. Dietze et al. (2022) show that the sub-daily as well as daily and multi-

day precipitation totals at the DWD weather station Weilerswist-Lommersum (Erft catchment) exceeded 500-year return period values. According to the KOSTRA classification of DWD (KOSTRA-DWD-2010R), 24 h totals also exceeded the 100-year return level widespread in the area between Cologne and the borders to Luxembourg and Belgium (see Fig. S3 including data description in the Supplementary material). Previous works have already shown that estimated return periods, which are much longer than the observational data, exhibit large uncertainty (Makkonen, 2006; Grieser et al., 2007). In particular, the extreme

high upper bounds of the estimated confidence intervals or the up to 50 % increased 100-year return level from KOSTRA-DWD-2010R indicate high statistical uncertainty in the return period estimation based on precipitation observations. In PART2 several thousand years of regional climate model simulations are used to reduce statistical uncertainty and to classify the event in a longer-term historical and projected climatological context.

### 3.1.2   Weather forecast analysis

In the following, the predictability of the event is analyzed based on weather forecasts by DWD and ECMWF (see Sect. 2.1.2). The deterministic forecast runs of the DWD ICON-EU model show the potential for a widespread heavy precipitation event in the border region between western Germany, eastern France, Belgium, Luxembourg, and Netherlands early as the 12 July 00:00 UTC (see Fig. S4 in the Supplementary material). While the affected area and intensity varies over the next forecasts, the potential for an extraordinary event in this region remains. Specifically for the affected area (LReg), high 24 h precipitation

totals (within the range of the observations; see Sect. 3.1.1) were predicted more than 2 days (from 12 July 00:00 UTC) ahead of the event (blue dots in Fig. 4a). The DWD ICON-D2 forecasts also captured the magnitude of the event right from of their first initialization (13 July 06:00 UTC; red dots in Fig. 4a), as can also be seen in the spatial distribution of the precipitation forecasts by ICON-D2 and ICON-D2-EPS for different initialization times (Fig. S5 in the Supplementary material). They consistently depict high precipitation totals over a wide area around the border between NRW, RP, Belgium, and Luxembourg,

locally exceeding 150 mm per day. Especially for the affected area, both ICON-EU and ICON-D2 forecasts oscillate around the observed precipitation totals (Fig. 4a), with even slightly higher predicted totals compared to the observations for the initialization shortly ahead of the start of the event. The DWD ICON-D2-EPS predictions (gray box plots in Fig. 4a) clearly


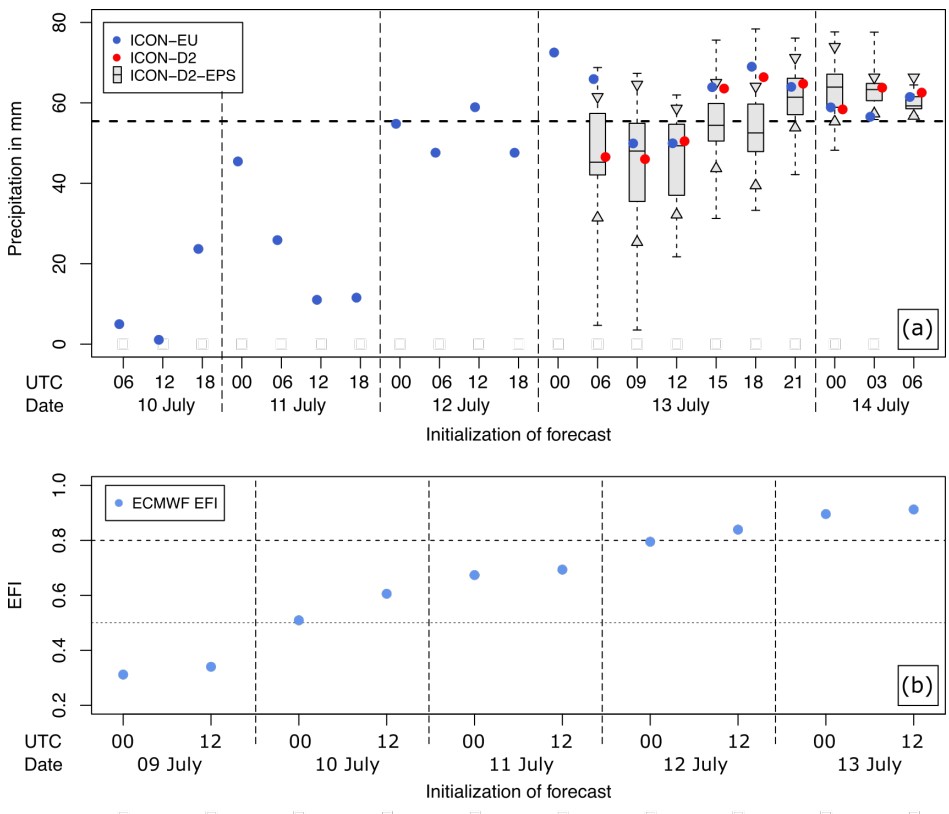

**Figure 4.** (a) 24 h precipitation totals over LReg (14 July 2021 06:00 UTC to 15 July 2021 06:00 UTC) as forecasted by ICON-EU, ICON-D2, and ICON-D2-EPS for different initialization times and observed precipitation total as reference based on RADOLAN: 55.4 mm (stippled black horizontal line). For the EPS, the boxes represent the median and 25 % or 75 % percentiles, the triangles the 10 % or 90 % percentiles and the whiskers the total ensemble range. (b) Extreme Forecast Index (EFI) for precipitation on 14 July 2021 based on the ECMWF-EPS for different initialization times. Horizontal lines at 0.5 and 0.8 denote the limits for classification of an unusual (EFI between 0.5 and 0.8) and very unusual or extreme event (EFI greater 0.8).

show that the closer they were initialized to the event, the more the uncertainty of the prediction decreased. The predictions of the ECMWF-EPS also indicated the possibility of an unusual event early on (Fig. 4b). EFI values (see Sect. 2.1.2) above 0.5 are

obtained from the forecasts initialized on 10 and 11 July 2021. From 12 July on, also 2 days before the event, the EFI for 24 h precipitation exceed values of 0.8, indicating a very high probability of occurrence of a very unusual or extreme event in this region. The above results show that while the precise prediction of the rainfall totals for the affected areas as only possible a few hours in advance, the potential for a extraordinary precipitation event in the region was given at least two days in advance.



## 3.2 Hydrological aspects

In this section, we discuss the hydrological aspects of the event, including antecedent conditions in the catchments, river water levels and streamflow, effects on reservoirs, and a comparison of observed peak values with statistical design floods. However, we did not estimate statistical return periods of the July 2021 flood for several reasons: The first is that during the event, many gauging stations were partly or completely destroyed, and even if water level recordings existed, water level-discharge relations ($W$-$Q$-relations) at many gauges were severely altered during the flood due to dynamical river bed changes or backwater

effects from floating debris trapped upstream of bridges (see Sect 3.3). Furthermore, the water levels observed during the flood were often all-time records exceeding existing $W$-$Q$-relations. Reconstructing event discharge, the basis for a statistical treatment, therefore is a difficult and still ongoing task, and the values reported below should be interpreted as the best available to date, but not as final estimates. Nevertheless, and despite considerable uncertainties, often the sheer magnitude of the estimates underline the exceptional nature of the event.


### Antecedent conditions

Analyzing major historical floods in Germany, Schröter et al. (2015) found that they were either triggered by extensive precipitation events following a normal to dry period, or moderate precipitation events following an exceptionally wet period. This means that catchment preconditions–namely soil wetness–are a crucial factor for flood occurrence and/or magnitude, which

we discuss in the following. We express the catchment preconditions by the API (Sect. 2.2.3) quantified from RADOLAN data (Sect. 2.1.1).

For the 30 days prior to the event, API (Fig. 3c) shows a wet period in most of Germany. The spatial averages in the study area are 69.6 mm for LReg and 74.0 mm for SReg, which is about twice the climatological mean (not shown). This agrees with Junghänel et al. (2021), who reported frequent rainfall in this region in the 3 weeks prior to the flood leading to widespread

high soil moisture. In the southern parts of the Eifel, the Ardennes in the north-west, and in the north-east of the study area in the Wupper region, generally less than 10 mm of soil water storage were still available for infiltration. In the remaining regions, free soil water storage was larger, but still below average, ranging mainly between 10 and 30, sometimes 75 mm. For Luxembourg, AGE (2021) also report relatively wet antecedent conditions.

In summary, high but not exceptional antecedent soil moisture was one factor supporting development of the observed floods

through saturation excess. In addition, the very high rainfall intensities (see Sect. 3.1.1) supported quick rainfall-runoff transformation through infiltration excess overland flow (cf. also AGE, 2021).

### Ahr, Kyll, and Prüm river basins

While draining into different directions, the Ahr, Kyll, and Prüm rivers all originate from the central part of the Eifel plateau

(see Fig. 1) and were hence exposed to the same rainfall in their headwaters. As a consequence, their flood dynamics were quite similar and we therefore discuss them together here. In their headwaters (Fig. 5a, c, e), with the onset of rainfall (see Sect. 3.1.1), water levels started rising in the morning of 14 July (08:00–10:00 UTC) and reached peak levels in the evening





(20:00–22:00 UTC). Within only 12 hours, water levels rose by about 3 m at Kyll and Prüm, and by more than 5 m at the Ahr. As all gauges measure water levels, the streamflow was calculated by $W$-$Q$-relations (carried out by the data-providing water authorities). Hence, there is uncertainty of about 15 to 20 % associated with the streamflow values shown in Figure 5 and Table 1. All peak values are clearly beyond the peak discharges of a 100-year flood event (HQ$_{100}$), for Müsch even by a factor of about 3 (Table 1). After the peak, water levels in the headwaters started gradually declining, and by around midnight of the following day (15 July) the flood had ended.

At the downstream gauges (Fig. 5b, d, f), peak water levels were reached in the early morning of 15 July (00:00 UTC at gauge Altenahr, 08:30 UTC at gauge Kordel, 05:30 UTC at gauge Prümzurlay). Like for the headwater gauges, water levels rose dramatically within only 12 hours: Almost 6 m at Kordel, almost 7 m at Prümzurlay, and almost 10 m at Altenahr. Likewise, the estimated peak discharges are all clearly above the statistical HQ$_{100}$ values, with a record factor of about 4 at gauge Altenahr (Table 1). Owing to its deeply incised topography, forcing settled areas into close vicinity of the river and leaving very little floodplains for safe inundation, the villages along the river Ahr were severely affected by the flood, including a large number of fatalities (see Sect. 4). Another consequence of the steep terrain and constricted conditions at the Ahr were high flow velocities, erosion and transport of floating debris, causing substantial blocking and backwater effects (see Sect. 3.3). Also, the gauge Altenahr–like many other gauges in the Ahr, Kyll, and Prüm river basins–was completely destroyed during the flood, such that the time series are partly reconstructions (done by the data-providing water authorities) based on floating debris lines at buildings, recordings of upstream gauges, and observed rainfall volumes (dashed time series in Fig. 5).

Owing to reconstructions, discharge estimates (not only) at gauge Altenahr include uncertainties of about 15 to 20 %. However, the estimated peak flow of about 900 to 1000 $\mathrm{m^3\,s^{-1}}$ (estimate by the the Water administration of RP) corresponds well with estimates of 1000 to 1200 $\mathrm{m^3\,s^{-1}}$ based on hydraulic considerations by Roggenkamp and Hergert (2022). Final estimates will be published in the *Deutsches Gewässerkundliches Jahrbuch* (www.dgj.de, last access: 9 May 2022) by the end of 2022. It is also noteworthy that while large historical floods (in 1804 and 1910) have been documented at the Ahr (Roggenkamp and Herget, 2014a, b), only data from the continuous records starting in 1973 were used to estimate HQ$_{100}$ values (see PART2 for a detailed discussion).

**Erft river basin**

Like the rivers Ahr, Kyll, and Prüm, the river Erft originates in the Eifel, but it drains northward towards its confluence with the Rhine near the city of Düsseldorf. In the Erft headwater region, 130 to 150 mm of rain fell on 14 July, with highest intensities occurring between 10:00 and 19:00 UTC. As a consequence, the water level at headwater gauge Schönau (Fig. 5g), for example, started rising at 07:00 UTC, reaching its peak at 18:50 UTC in the evening, more than 5 times larger than the statistical HQ$_{100}$ of the gauge. At gauge Bliesheim (Fig. 5h), 36 km downstream of Schönau, water levels started rising about 6 hours later, but–due to the operation of the retention basins Eicherscheid and Horchheim located in-between–reached its peak only at 08:45 UTC on the following day. While the operation of the reservoirs delayed the peak somewhat, they were far too small to significantly reduce the flood peak: At Bliesheim, the maximum discharge exceeded the statistical HQ$_{100}$ by a factor of more than 7 (Table 1), which is even higher than for Schönau.





**Figure 5.** Time series of water level (blue) and streamflow (red) for river gauges in the Ahr, Kyll, Prüm, Erft, and Wupper river basins (13 July 2021 00:00 UTC to 17 July 2021 23:00 UTC; resolution: 15 min). Solid lines indicate values based on in-situ measurements, dashed lines indicate reconstructed values (carried out by the data-providing water authorities). Dashed horizontal lines indicate the magnitude of the statistical 100-year flood in m³ s⁻¹. Further details about the gauges are given in Table 1.





**Table 1.** Key characteristics of river basins and gauges (Water level $W$; streamflow $Q$; $HQ_{100}$ means a flood with statistical 100-year return period; peak factor is defined as $\max(Q)$ in 2021 divided by $HQ_{100}$) including statistics of previous historical extremes and the July 2021 flood event. Values for the latter are approximations (estimated).

| Gauge name | Basin size (km²) | Measuring period (years) | Previous historical extreme | | | Statistical extreme | Flood event in July 2021 | | | |
|---|---|---|---|---|---|---|---|---|---|---|
| | | | Date | Max. $W$ (cm) | Max. $Q$ (m³s⁻¹) | $HQ_{100}$ (m³s⁻¹) | Peak time (UTC) | Max. $W$ (cm) | Max. $Q$ (m³s⁻¹) | Peak factor (–) |
| Ahr river basin[a] | | | | | | | | | | |
| Müsch | 353.2 | 1973–2019 | 02 June 2016 | 273 | 132 | 152 | 14 July 20:15 | ca. 650 | ca. 500 | ca. 3.3 |
| Altenahr | 749.0 | 1946–2019 | 02 Feb. 2016 | 371 | 236 | 241 | 15 July 00:00 | 984–1019 | ca. 1000 | ca. 4 |
| Kyll river basin[a] | | | | | | | | | | |
| Jünkerath | 175.6 | 1973–2019 | 17 Dec. 1974 | 266 | 129 | 118 | 14 July 22:45 | ca. 370 | ca. 200 | ca. 1.7 |
| Kordel | 816.3 | 1968–2019 | 26 Jan. 1995 | 481 | 218 | 248 | 15 July 08:30 | ca. 600 | ca. 600 | ca. 2.5 |
| Prüm river basin[a] | | | | | | | | | | |
| Prüm 2 | 53.2 | 1976–2019 | 07 Feb. 1984 | 126 | 43.5 | 51.6 | 14 July 21:15 | ca. 330 | ca. 120 | ca. 2.3 |
| Prümzurlay | 576.1 | 1973–2019 | 03 Jan. 2003 | 492 | 252 | 278 | 15 July 05:30 | ca. 700 | ca. 600 | ca. 2 |
| Erft river basin[b] | | | | | | | | | | |
| Schönau | 31.9 | 1972–2020 | 16 Mar. 1988 | 129 | 17.5 | 19 | 14 July 18:50 | ca. 200 | ca. 100 | ca. 5.2 |
| Bliesheim | 604.2 | 1965–2020 | 31 May 1984 | 247 | 56.2 | 71 | 15 July 08:45 | ca. 400 | ca. 500 | ca. 7.3 |
| Wupper river basin[c] | | | | | | | | | | |
| Hückeswagen | 163.2 | 1987–2020 | 28 Dec. 1994 | 272 | 64 | 73 | 14 July 21:40 | ca. 430 | ca. 200 | ca. 2.9 |
| Opladen | 606 | 1950–2020 | 23 Sept. 1957 | 306 | 219 | 250 | 15 July 02:45 | ca. 460 | ca. 530 | ca. 2.1 |

[a] Operator and data provider: Water administration of Rhineland-Palatinate (www.lfu.rlp.de, last access: 9 May 2022)

[b] Operator and data provider: Erftverband (www.erftverband.de, last access: 9 May 2022)

[c] Operator and data provider: Wupperverband (www.wupperverband.de, last access: 9 May 2022)

In fact, the magnitude of the flood not only rendered flood reduction by reservoir operation impossible, it even posed a great threat to many retention basins in the study area. As a typical example, we briefly summarize the course of events at retention
basin Horchheim (see Fig. 1). It was built in the 1980s and is operated by the Erftverband for downstream flood protection. The reservoir volume and outlet gates are designed for protection from $HQ_{100} = 58\,\mathrm{m^3\,s^{-1}}$, the design flood for ensuring dam stability is $HQ_{10\,000} = 90\,\mathrm{m^3\,s^{-1}}$. In the night from 14 to 15 July, water entering the reservoir exceeded any recorded values. Between 02:00 and 05:00 UTC in the morning of 15 July, i.e., in only 3 hours, total inflow exceeded twice the entire retention volume. The peak inflow was estimated as about $390\,\mathrm{m^3\,s^{-1}}$, which is about 4 times the $HQ_{10\,000}$. As a consequence, and
despite all flood gates opened, the dam overtopped at 05:35 UTC, causing destruction of all service infrastructure and partial destruction of the dam by backward erosion (Erftverband, 2021). Similar situations were–among many others–encountered at retention basins Eicherscheid and Niederberg and service water reservoir Steinbachtalsperre (all located in the Erft basin) and Urfttalsperre (river Urft, see Fig. 1). For the latter, the design discharge for dam stability of $HQ_{10\,000} = 289\,\mathrm{m^3\,s^{-1}}$ was





surpassed by a factor of more than about 1 in the morning of 15 July (estimated peak discharge is about $320\,\mathrm{m}^3\,\mathrm{s}^{-1}$), fortunately
without dam failure (Bung, 2021).

Since most gauges along the river Erft–among them Schönau and Bliesheim–were heavily bypassed during the flood, and because water levels exceeded any recorded levels, existing $W$-$Q$-relations were often not adequate to estimate peak discharge. For gauge Schönau, peak flow was estimated from rates of change of basin volume of the flood retention basin Eicherscheid, which is located just 1.6 km downstream of the gauge. For gauge Bliesheim, peak flow was estimated by Erftverband using
a two-dimensional (2D) hydraulic model (Hydro_as-2d) reproducing observed water levels. For the model, feasible ranges of roughness coefficients suggest an uncertainty range for peak discharge of $\pm\,100\,\mathrm{m}^3\,\mathrm{s}^{-1}$.

**Wupper river basin**

Unlike the previously discussed river basins, the Wupper river basin is located east of the Rhine (see Fig. 1). It is charac-
terized by low mountain terrain and several reservoirs, most of them operated by the Wupperverband. The largest reservoir is the Wupper-Talsperre. It has an upstream basin size of $212\,\mathrm{km}^2$, an overall storage volume of $25.6 \cdot 10^6\,\mathrm{m}^3$, additionally $9.9 \cdot 10^6\,\mathrm{m}^3$ are available for flood retention. Gauge Hückeswagen (Table 1 and Fig. 5i) is just upstream of the reservoir, gauge Opladen (Table 1 and Fig. 5j) is far downstream, close to the confluence with the Rhine (see Fig. 1). Just like in the Eifel region west of the Rhine, the rainfall event in the Wupper basin on 14 July was characterized by high intensities, large sums, and
large spatial extent (see Fig. 3a): In an area larger than $1000\,\mathrm{km}^2$, 120 to 150 mm of rainfall were recorded within 24 hours on 14 July. Even compared to point statistics, these totals exceed 200-year return intervals (see Fig. 3b), in combination with its areal extent the return period is even more extreme and hard to quantify (cf. PART2). As a consequence, water levels and discharge at gauge Hückeswagen rose to magnitudes almost 3 times larger than the previously largest recorded flood and the statistical 100-year flood (Table 1). At about 15:00 UTC, the gauge was completely inundated and data transfer stopped. Thanks
to warnings by DWD, the Wupper-Talsperre water level was drawn down at increasing rates since 12 July, creating additional retention volume prior to the flood. Despite these measures, its maximum water level was reached and exceeded at 22:00 UTC on 14 July, and the flood spillway was activated to prevent dam failure. Overall, and largely thanks to the pre-event water level drawdown, the reservoir fulfilled its protection task by reducing the unprecedented maximum inflow of about $230\,\mathrm{m}^3\,\mathrm{s}^{-1}$ to about $190\,\mathrm{m}^3\,\mathrm{s}^{-1}$ downstream (Wupperverband, 2021). At the Wupper-Talsperre, like for the previously mentioned reservoirs
Horchheim and Urfttalsperre, the maximum inflow exceeded even the $\mathrm{HQ}_{10\,000} = 168\,\mathrm{m}^3\,\mathrm{s}^{-1}$ design discharge, here by a factor of about 1.3.

Despite the mitigating effect of the reservoir operation, and largely owing to the effect of the large intermediate catchment area, peak water levels and streamflow at downstream gauge Opladen also were the largest ever recorded, exceeding the statistical 100-year flood by a factor of more than 2 (Table 1). This is underlined by the fact that the recorded peak water level of
about 466 cm was well beyond the limit of the existing $W$-$Q$-relation of 400 cm; streamflow values beyond were reconstructed by Wupperverband using a 2D hydraulic model (Hydro_as-2d; Fig. 5j). During the flood, the river cross-section at Opladen was excavated by 40 to 50 cm, making the flood not only a hydrological but also a morphological event (see also Sect. 3.3).

**Rivers in Belgium and Luxembourg**

Flooding in Belgium occurred mainly in the south-eastern parts, on tributaries to the river Meuse originating from the Eifel and Ardennes region (see Fig. 1). Especially along the rivers Ourthe (basin size 1850 km$^2$), Amblève (1100 km$^2$), and Vesdre (700 km$^2$), previously unobserved peak water levels and streamflow values were reached, leading to 39 fatalities and vast damage to buildings and infrastructure (Dewals et al., 2021). Rainfall amounts close to 200 mm in 24 hours were observed in the region, which is far beyond the statistical 200-year return period (Dewals et al., 2021). Similar to the German parts of the

Eifel, the main characteristic of the rainfall event was a long persistence of high intensities over a large area, leading to a very fast rise of river water levels to unprecedented heights. For example, peak discharge at the Ourthe river just upstream the Vesdre confluence was estimated to be about 1100 m$^3$ s$^{-1}$, which is about 50 % above the previously observed maximum and about 25 % above the statistical 100-year flood. Along other rivers such as the Vesdre, evidence from partial gauge observations (most were destroyed during the flood) suggests that 100-year floods were exceeded by a factor of about 3 (Dewals et al., 2021).

In Luxembourg, severe flooding occurred mainly along the rivers Alzette, Sur, Our, and tributaries in the east and south-east of the Grand Duchy (see Fig. 1). In the most affected regions, the related 24 h precipitation totals reached 100 mm. At rain gauge Godbrange, it even amounted to 105.8 mm (14 July 05:00 UTC to 15 July 05:00 UTC), the highest recorded 24 h precipitation total in Luxembourg since the beginning of recordings in 1851 (MeteoLux, 2021). At ten river gauges, streamflow surpassed the statistical HQ$_{100}$, at 15 gauges it was the highest ever recorded, resulting in widespread inundations and evacuation (AGE,

455 2021).

### 3.3 Hydro-morphodynamic processes

**Fluvial morphology as a dynamic system**

A river system is dynamic and non-linear interactions occur between the water phase, fixed and erodible boundaries, moving sediment and moving debris, and vegetation (Yalin, 2015). These non-linear interactions ultimately shape the morphology

of the river channel networks with impact on flow hydrodynamics and ultimately on flood hazard due to the change in the flow boundaries (cf. Dietze et al., 2022, for a landscape perspective of the July 2021 floods addressing some of these issues). Flood-induced alterations in the morphology of landscape, river channels, and urban spaces have large potential to destruct and damage housing, farm land, industry, crucial infrastructure, and natural areas (see also Sect. 4). All this was observed during the July 2021 flood in Germany (cf. also Dewals et al., 2021), which provides a showcase of hydro-morphodynamic processes

conditioning the valley response to such extreme events, as well as how changes in the valley morphology occurring during such events exacerbate the flood hazard.

In the municipality of Rech (Ahrweiler district), for instance, the erosion and collapse of the right bank of the river Ahr destructed houses and infrastructure (cf. cover photo at Petermann, A., 2021, "*Um Dernau herum wurde die ganze Talsohle zugebaut–Futter für die Flut, die im Juli kam.*"). Another very visible morphological effect was the destruction of gauging

stations (e.g., the destruction of the gauging station Altenahr; see Sect. 3.2) by local scour, bank erosion and collapse, which hindered the hydrological monitoring and reconstruction of the event.

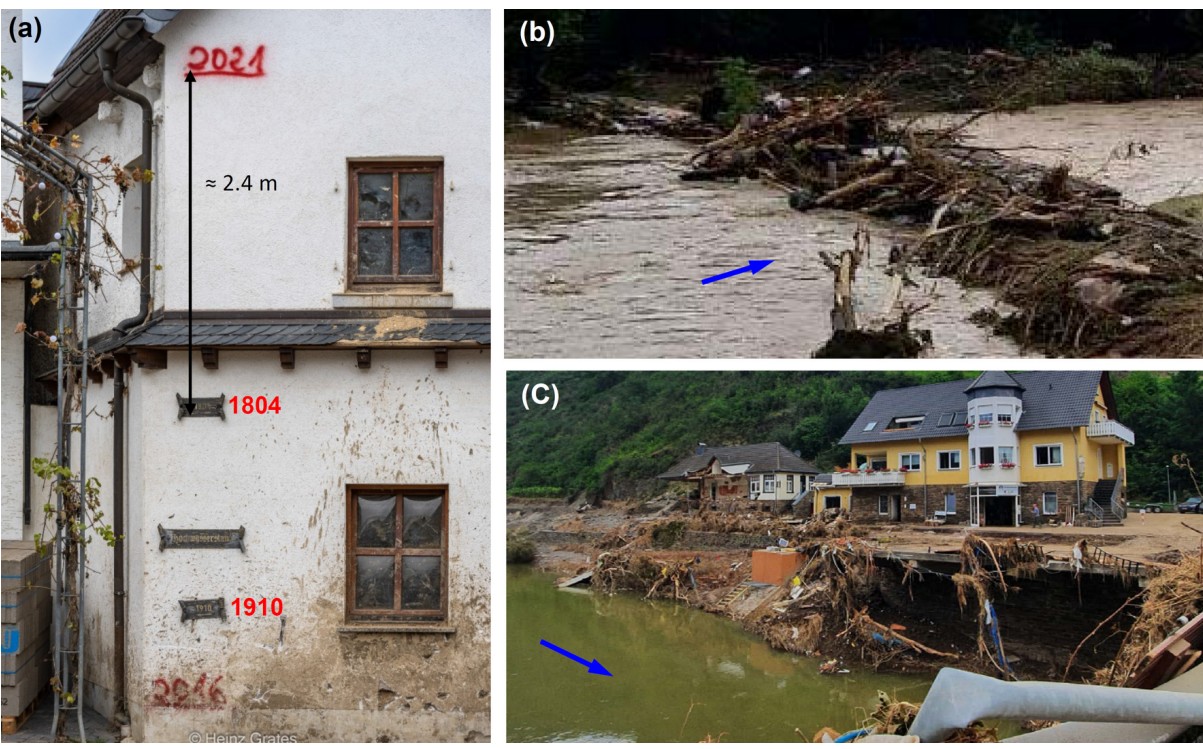

**Figure 6.** (a) Flood marks in municipality of Dernau (Ahrweiler district), including the floods of 1804, 1910, 2016, and 2021 (© Heinz Grates). (b) One of the collapsed bridges of the Ahr valley railroad (Ahrtalbahn) with trees eroded from the landscape (© Martin Seifert) and (c) bank erosion and collapsed road bridge, both in the municipality of Altenahr (Altenburg; © Bettina Vier). Blue arrows show the river flow direction.

A clear example of the dynamic character of a river valley is the municipality of Dernau (also in the Ahrweiler district): While the peak flood discharge estimated for the July 2021 flood was in the same order of magnitude as for the 1804 event (cf. PART2 and Roggenkamp and Hergert, 2022), the flood level in 2021 was about 2.4 m higher (Fig. 6a). The explanation

of this outstanding difference is not straightforward, but can be partially explained by hydro-morphodynamic considerations. Dernau is located on a relatively wide section of the Ahr Valley, which permitted urbanization to develop on a floodplain of the compound river cross-section. Immediately downstream, the river becomes single-threaded and narrow for approximately 2 km, widening again just upstream of the town of Bad Neuenahr-Ahrweiler. This topography, with a downstream bottleneck, which can be seen as an internal river control section, makes Dernau naturally prone to water accumulation and consequent

inundation due to backwater effects. Between 1804 and 2021, more precisely in the 1880s, the railroad of the Ahr Valley (Ahrtalbahn) was constructed and four bridges were built precisely in the bottleneck river reach. This reduced even further the cross-section available to the flow on an already naturally narrow reach of the river. During the July 2021 flood, these bridges were destroyed along with the Ahr Valley railroad (Fig. 6b).



From the post-event analysis, we may argue that the higher peak water level–compared to 1804–were due to a combination
of both: (a) the anthropogenic morphology change of the valley geometry, more specifically of the river cross section, imposed
by the construction of the railroad bridges, which increased the bottleneck effect; and (b) the probable damming and clogging
of the river cross sections at the location of the bridges by the trapping of large-scale debris transported from the upstream val-
ley (in Fig. 6b) remains of this debris, mainly wood, are visible). Another plausible contribution, which is however impossible
to corroborate, may be related to the actual existence of a higher volume of large-scale debris available to be picked up in the
upstream valley. This includes wood, which can be naturally recruited from hillslopes and valley bottom (Comiti et al., 2016),
and also a considerable amount of industrial components nonexistent or limited in 1804, such as vehicles and caravans, bins
and containers, and construction materials.

**More than water: sediment and debris**

Hillslope denudation and widespread landslides are the most evident impact of extreme hydro-meteorological events in the
upstream regions of a catchment. While these impacts may happen far away from the river network, they nevertheless provide
an excess of sediment into the river channels, which in turn contributes to changes in river morphology, interacting non-linearly
with the flood propagation and leaving a trace in the pathway of the flow. This happened to a large extent during the July 2021
event. Additionally, organic and inorganic large-scale debris was recruited from urban, rural, and natural landscape regions,
often with dimensions comparable to the cross-section available to the flow (cf. Twitter-Tweet from NoeWehrtSich, 2021).
Consequently, river sections and streets were partially and sometimes entirely dammed (Dewals et al., 2021).

Assuming that the transport of large debris follows the same modes as described in Ruiz-Villanueva et al. (2019) for large
wood, the post-events images taken at the river Ahr indicate that, at least for some moments during the event, the large debris
was transported in the so-called congested and hypercongested regimes, in which the debris elements are constantly in contact
and are transported as a continuous carpet at the surface of the flow. The presence of large debris under transport, in any of
these modes, has consequences to the flood propagation and impact: (a) offering extra resistance to the flow hence enhancing
flood levels; (b) blocking partially or completely the cross section available to the flow; (c) provoking destruction of buildings
and infrastructure by impact; (d) injuring (often deadly) people and animals; (e) destructing farm land and natural and cultural
protected sites; (f) and probably changing the flow properties (rheology) with consequences for modeling approaches. The
feedback between large-scale debris and fluvial geomorphology is still an open question with only a few existing studies for
the impact of large wood in rivers (Ravazzolo et al., 2020), which should, however, be evaluated further. These are all aspects
that are not considered in the current practice of flood hazard modeling, underestimating the real risk (Sect. 4.2).

**Natural and anthropogenic landscape singularities**

In a river catchment, many singularities disrupt the continuity of the landscape and river network. These can be natural such as
rock protrusions from the river bed or banks, or anthropogenic such as bridges, tunnels, or mining pits, to name a few examples
(Dietze et al., 2022, refers to the legacy of human use, which is imprinted in the landscape). These landscape singularities
are responsible for: (a) flow separation with possible erosion enhancement; (b) acceleration and deceleration flow regions; (c)



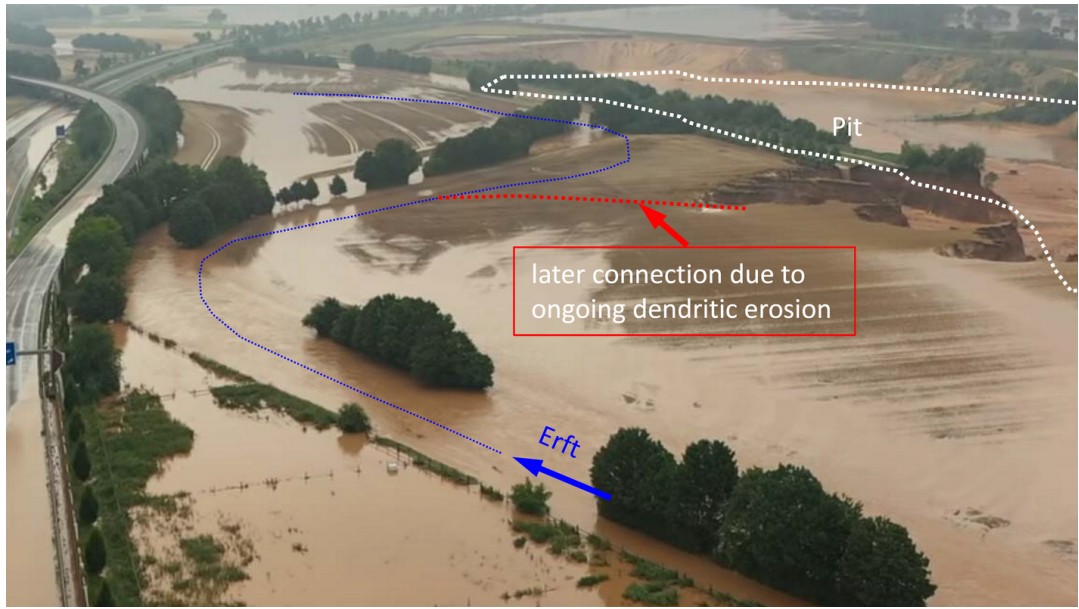

**Figure 7.** View of pit Blessem (belongs to Erftstadt) during the flood with the location of the later connection of the river Erft to the pit (Maurice, 2021). Blue arrow shows the river flow direction.

water accumulation and/or sediment deposition; (d) extra resistance to the flow; (e) morphology changes due to unaccounted

fragility of the landscape; and (f) bypass possibilities.

The most visible anthropogenic singularities in valleys, which influence and are affected by floods, are bridges. Fekete and Sandholz (2021) refer to the destruction of 62 bridges, whereas BMI (2022) estimated that 103 bridges were damaged or completely destroyed during the July 2021 flood event, only in the Ahr Valley, and it is known that extreme floods is one common reason for bridge collapse (Deng et al., 2016). Several reasons can be pointed out for the bridge collapse of two

examples in the Ahrweiler district (see Fig. 6b and c): foundation excavation and scouring, impact and hydrodynamic action, lateral erosion of the abutments, and bank failure.

Ephemeral morphology features such as channel section blockage are provoked by accumulation of sediments and debris, and these occurred at several locations during the July 2021 flood. The eventual destruction and collapse of these impounding incidents and related upstream temporary reservoirs caused downstream flood bursts (Fekete and Sandholz, 2021) with extra

debris-charged peak discharges traveling downstream as an abrupt front (with similar effects to dam break flows).

An (anthropogenic) landscape fragility in the Erft catchment had unforeseen consequences (and excessive media exposure) on the valley security and on the flood impact: a mining pit as deep as 50 m and with an area of roughly 25 ha, located northwest of the town of Erftstadt (Erft catchment, Rhein-Erft-Kreis district). During the beginning of the 15 July, the urban district of Blessem of the town of Erftstadt was inundated and the flood flowed freely through the streets, roughly from south to north.

The pathway of the water downstream the town was along the natural landscape gradient and eventually into the pit through an





existing depression on its south levee. This flow pattern corresponded remarkably to what the simulations from the flood hazard map produced by the district government of Cologne previewed for both cases: a flood with a return period of 100 years and an extreme flood (Bezirksregierung Köln, 2019). When the water started flowing on the pit slope, an unexpected major process of retro dendritic (tree-shaped) erosion occurred, which reached the urban district of Blessem in 6 to 8 hours (Fig. 7). Eight houses

were destroyed and considerable damage was reported to infrastructure and other buildings and assets. During the night of 15 to 16 July, the dendritic shape of the erosion reached the river Erft and redirected it into the pit. This dramatic and unforeseen morphological incident drastically changed the configuration of the landscape within only 1.5 days. Although the possibility of drainage of the flood into the mining pit was known, nobody expected the erosive process it would trigger. Interestingly, also a positive side effect was observed once the river Erft was connected to the mining pit: the flood volume was partially absorbed

by the reservoir formed by the pit, with mitigating effects for the downstream valley. This positive experience could be used to integrate the numerous mining pits that exist in the Rhine valley in a flood management system.

Other singularities, often overlooked in flood risk management, are the bypass possibilities in the landscape. However, they may cause serious consequences when extreme events occur, either in changes in the valleys morphology, or by enhancing the downstream floods. Such was the case of the road tunnel in the municipality of Altenahr constructed in 1834. When the flood

in July 2021 reached a level higher than the western entrance of the road tunnel of the federal highway B267, the water started flowing freely through it, bypassing a 3 km meander of the river Ahr. Meanders generally contribute to peak discharge attenuation (Buffin-Bélanger et al., 2015), so in this particular case the bypass resulted in an increase in the downstream discharge. Considerable destruction was caused at the eastern tunnel outlet, where massive scouring of the road and the neighboring slope occurred. A scour erosion step of more than 4 m was created, which the water passed as a water fall. Furthermore, the bypassed

flow conditioned the flow in the downstream river, influencing its morphology and creating new areas of deposition and erosion.

**3D hydrodynamic complex processes**

Common conventional flood analyses are performed with 1D or 2D depth averaged models for clear water, which do not capture the complex 3D hydrodynamic processes that can be crucial for flood propagation and its consequences (Bates, 2022).

Such were the cases in the municipality of Schuld and urban district of Altenburg (municipality of Altenahr), both located in a meandering region of the river Ahr. The southern part of the urban settlement of Schuld is located on the left hand side of the inner region of an Ahr bend. It is well known, theoretical and empirically, that the inner region of a river bend is a deposition area (cf. Blanckaert and de Vriend, 2003). During the July 2021 flood, the flow inundated this area of the urban settlement. The water, charged with sediment and debris, circulated through the streets and, as shown in Figure 8, the solid material deposited,

with serious consequences for the security of the dwellers and buildings.

In the second case, the water over-topped the levee of the river Ahr and was directed towards an old meander on the right hand side riverbank, flooding the southeast of the urban district of Altenburg (see Sect. 4.1 Fig. 10). The spreading of the flow through what possibly is an abandoned oxbow lake (cf. Dey, 2014, on the formation of oxbow lakes, which are abandoned meanders not linked to the main channel anymore), may have contributed to an attenuation of the downstream peak discharge





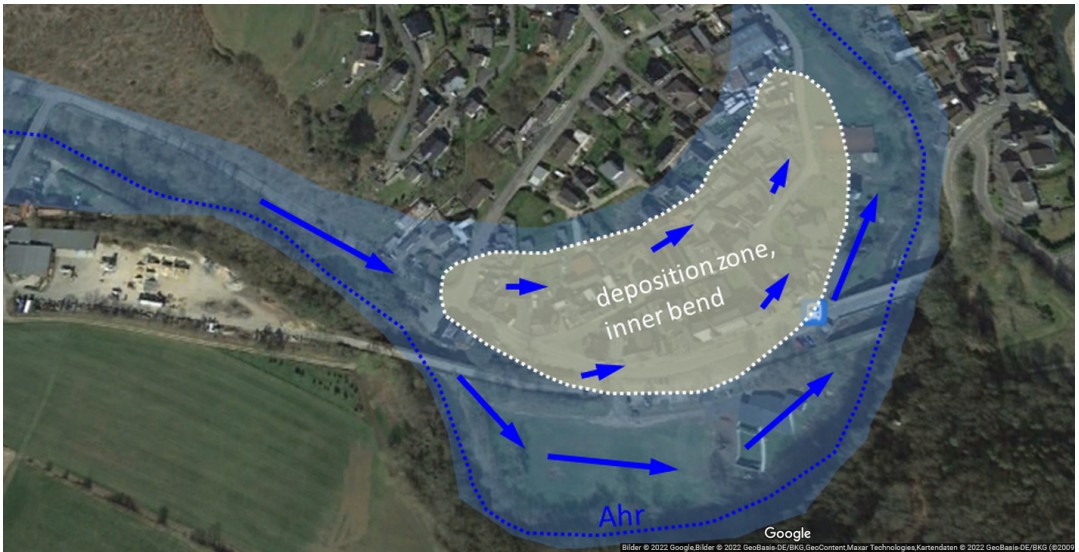

**Figure 8.** Inner bend deposition in the municipality of Schuld (© Google Earth, 2021).

through the flood lamination effect. However, oxbow regions are preferential areas for the deposition of sediment and debris
due to reduced flow velocities, which in the case of urban district of Altenburg has destructive effects in the urban area.

The interaction of the flowing water with the landscape and rivers morphology enhances the destructive power of a flood.
Some examples of this enhancement of the impact and consequences of such major events include bank and infrastructures
collapse by scouring, introduction of flood bursts due to cross-section damming and sudden break, and collision with buildings
and dwellers.

## 4   Impacts and consequences

The July 2021 flood event caused severe damage to buildings and infrastructure in several German districts in the federal states
NRW and RP. In total, at least 180 people lost their lives (see Table S2 in the Supplementary material), 69 of them in the
Ahrweiler district (RP) along the river Ahr, and over 800 people were seriously injured (BMI, 2022). The Ahr Valley was the
hardest affected location, with an estimated 17 000 out of 42 000 people losing most of their property (BMI, 2022). Due to
gaps in the information warning chains (cf. Fekete and Sandholz, 2021; Thieken et al., 2022) and no widespread evacuations,
many people surprised by the fast rising floodwaters were unable to get to safe places or underestimated the imminent danger,
respectively (e.g., attempting to save belongings from the basement).

While the flood was ongoing and during the first days and weeks thereafter, the overall situation on site was relatively
uncertain, which posed a major problem for crisis management, emergency personnel, and the provision of relief supplies. As
part of CEDIM's concept of FDA (see Sect. 1), a first rapid assessment of the situation on site was carried out within a few days





after the event (cf. Schäfer et al., 2021) to obtain initial estimates of the inundation areas (Sect. 4.1), the associated potential losses (Sect. 4.2), and further consequences such as rail and road damage and blocking (Sect. 4.3).

## 4.1 Estimation of inundation areas in a rapid context

One of the first steps in rapid disaster analyses is to estimate the event's footprints and to identify affected areas. In case of the July 2021 flood, the identification of the inundation area was key. Even while flood was still ongoing at the time of CEDIM's inundation area estimation, live footage and imagery coming up on social media (e.g., Twitter, Facebook) were collected manually and used for the analysis to provide a rapid overview of the overall situation. Some news media broadcasted from helicopters flying over the region and provided aerial imagery. In addition, private videos from drone pilots and uploaded, for

example, to YouTube, complemented the footage analysis. With the understanding of the affected regions (e.g., orography, building and capital stock) and a first damage assessment using available loss models in CEDIM (Sect. 4.2), the groundwork for a more detailed FDA was provided and made available to the community (Schäfer et al., 2021; KIT, 2021).

The flood along the river Ahr lasted for about 3 days. The onset of the event started in the late hours of 14 July and lasted until 16 July (see Fig. 5). Throughout this period, information about the extent of the flooding was mostly generated by news

and social media. Initially, it was unclear which towns and villages were flooded and to what extent, and how severe the flooding might be. This was not only the case for the Ahr Valley, but also for most other affected regions such as along the river Erft. River gauge stations only provided limited information about the actual water level (destroyed gauges, see Sect. 3.2). Each collected photo and video was georeferenced and utilized to create geospatial vertices of the inundation areas. In many cases, the correct location, where a photo was taken or the flight path of the helicopter or drone, had to be determined first. By

comparing the photo and video footage with satellite imagery acquired in previous years, such as Google Earth, it was possible to roughly estimate the inundation areas. Simple landmarks such as garden fences or buildings helped in this effort. Figure 9 provides an overview of the final product for the Ahr Valley, where about 17 km$^2$ of flooded area was mapped between the two municipalities of Antweiler and Sinzig (distance around 34 km), excluding areas for which no image data were available.

In addition to the mentioned material above, Sentinel-1 imagery (see Sect. 2.1.4) helped to identify further flooded areas.

Furthermore, automatic procedures by the German Aerospace Center (Deutsches Zentrum für Luft- und Raumfahrt; DLR, 2021) were also integrated into the analyses, which were made available on 15 July–but without further review or filtering. Indeed, one constraint is that Sentinel-1's automated procedure is sensitive to topography and, for example, often identified hillslopes as inundation areas. Nonetheless, the information was an additional proxy to identify flooded areas, especially for regions without available photos or videos. News media focused mainly on severely affected places such as the town of Bad

Neuenahr-Ahrweiler or the municipality of Insul and ignored many other villages in between. In some cases, this gap was closed by private footage, but this was not always the case.

On 17 July, Sentinel-2 imagery was made available, allowing further flooded areas to be identified despite some cloud cover, and confirming previous findings. However, due to the low resolution of the imagery, manual assessment was necessary, for example by comparing the imagery with topography data to differentiate between flooded and muddy area. One problem was

that the mud extended to roads due to traffic and rain that were not flooded. In addition, some muddy roads were also caused

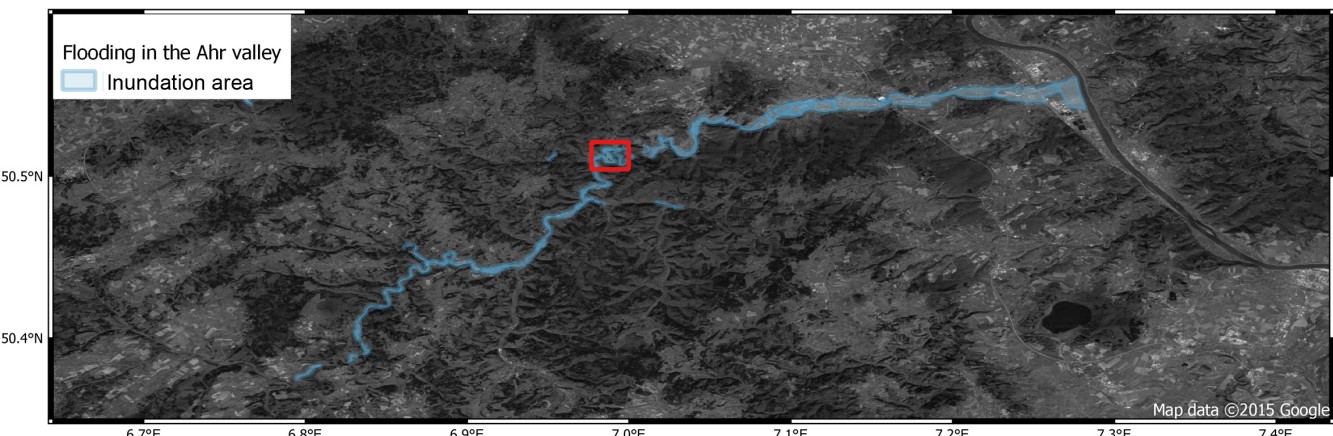

**Figure 9.** Overview of the inundation area along the river Ahr at the time of largest extent (blue), based on various sources including private drone footage, photos from social media, and satellite imagery (as of 18 July 2021). The red box indicates the area shown in Figure 10 (© Google Earth).

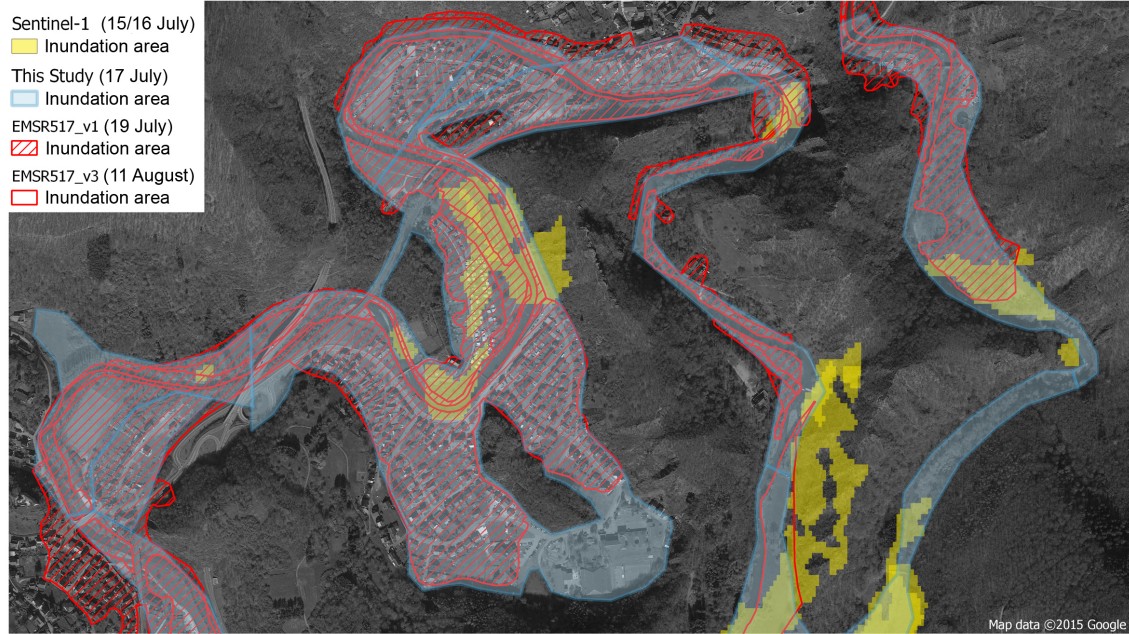

**Figure 10.** Comparison of the inundation area (municipality of Altenahr): Based on our method (in blue) from 17 July, results based on Sentinel-1 (in yellow; DLR, 2021) from 15/16 July, and results of the Copernicus Emergency Management Service (in red; COPERNICUS EMS, 2021) from 19 July (EMSR517_v1) and 11 August (EMSR517_v3; © Google Earth). For a photo comparison of the river loop before and during the flood see also tweet on Twitter: https://twitter.com/WxNB_/status/1415629704472760324 (last access: 9 May 2022).





by downhill water streams. Figure 10 compares exemplary different assessment products of the inundation area for the river loop near the municipality of Altenahr. For this specific region near Altenahr, photos highly supported the reconstruction of the inundation area (e.g., see tweet on Twitter: https://twitter.com/WxNB_/status/1415629704472760324, last access: 9 May 2022). It can be seen that purely satellite-based products (in yellow) or other rapid flood maps (in red; Copernicus Emergency

Management Service; COPERNICUS EMS, 2021) are similar but were not able to cover the inundation area in full detail.

Rapid estimation of the inundation area is crucial to quickly estimate impacts and losses. Automatic solutions as well as single data source products have not shown to be sufficiently accurate. To increase accuracy, it is highly recommend to use any data source available, especially including private photos released on social media platforms. In all cases, expert review is required to interpret the available data. This tedious work was the foundation for the loss modeling in the next section.

**4.2    Rapid loss estimation and further loss statistics**

Based on the rapid quantification of the mapped inundation areas in Sect. 4.1, loss models available in CEDIM (e.g., Daniell et al., 2011, 2018; Mühr et al., 2017), and empirical data from past flood disasters (hazard information, infrastructural and other damage), a first loss estimation immediately after the flood (within 1 week) was performed (Schäfer et al., 2021). A central element for this assessment was a very large natural disaster database with over 60 000 entries (CATDAT; Daniell et al.,

2011, 2016; EEA, 2022) built up by CEDIM in recent years.

The following characteristics and assumptions were made as part of the damage modeling: (a) based on the inundation areas, 9694 buildings in RP and 9702 buildings in NRW were identified to be affected; (b) based on the capital stock model used in Wilhelm et al. (2021), a capital stock of EUR 8.92 billion (excluding infrastructure) was associated with these buildings.

For the next points, a more in-depth description is provided: (c) the damage ratio defined within the flooded locations of the

mapped inundation areas ranged between 15 % and 33 % (economic damage as a % of capital stock). This was derived from previous analyses of past flood events, showing, for example, for the Ahr catchment that more buildings were generally located in areas with shallow inundation depths than in areas with high inundation depths. The damage ratio range was derived from the proportional integration of various vulnerability curves with building stock data of the most affected areas in NRW and RP assuming a typical building style of 2-3-story buildings.

In general, it is known that the extent and depth of floods primarily derived from satellite data is often underestimated and, thus, underestimate the number of buildings affected (and thus the damage ratio or loss). The main reason is that the peak of the hydrograph of the flood is not at the same time as the satellite passes. In addition, the assessments often concentrate on the most severely affected locations rather than on all locations. Based on the information mentioned above (especially the identified inundation areas from Sect. 4.1), we estimated a total building damage of about EUR 2.82 billion in these zones.

However, it should also be taken into account that the mapped inundation areas only contain a sample of the total flooded locations across Germany in the flood sequence. In addition, the narrowness and steepness of the affected valleys meant that accurate mapping of the peak flow was not possible by remote sensing methods. Further, subsequent damage surveys revealed that the flood heights were higher than the heights expected at the time. That means that a slightly higher average damage ratio over the entire mapped inundation areas should have been used.


Other studies from the summer 2021 reported a significantly higher number of affected buildings in the study area (or in Germany in general) compared to our estimates, such as the flood product from ICEYE, a Finnish microsatellite manufacturer that uses synthetic-aperture radar (SAR) satellite data to conduct flood monitoring (ICEYE, 2021). This is also the case according to reports from the Gesamtverband der Deutschen Versicherungswirtschaft (GDV), who registered 190 000 claims for the whole of Germany, of which 160 000 were residential and 30 000 commercial ones (Baker, 2021). 135 000 of these claims

were in NRW, 33 000 in RP and the remaining 20 000 in Bavaria and Saxony among other locations (also affected by the low-pressure system *Bernd*). Besides the fact that these studies also took into account other regions in Germany, another reason for the discrepancy to our estimates is that in the GDV report, for example, it is unclear which percentage of damage was related to rain intrusion or flooded properties. From this, it can be seen that our mapped inundation areas represented only a small portion of the total flooded stock. For this reason, we applied in a further step a scaling factor to account for this discrepancy.

Given the unknown number of affected buildings and infrastructure outside of the estimated footprints from Sect. 4.1, scaling factors ranging between 3.4 and 8.1 were used to estimate the share of damage in these regions. These scaling factors represent the additional exposure vs. that of the mapped inundation areas. However, as is generally the case, the most affected areas were already included in our mapped inundation areas, thus a lower damage ratio compared to the first estimation was to be expected after applying the scaling factors. This approach is in line with existing studies under the World Bank's Global Rapid

post-disaster Damage Estimation (GRADE) methodology (Gunasekera et al., 2018)[1] when estimating damage in locations where absolute inundation depths are unknown.

Thus, in the frame of CEDIM's FDA 1 week after the event, the following estimations were provided, which apply to the whole of Germany–note, however, that the proportion for Saxony and Bavaria is only about 1 %: (a) Damage to private assets (including household goods): EUR 4.4–13.0 billion; (b) damage to commercial, industrial, and other buildings: EUR 1.8–3.9

billion; (c) damage to infrastructure: EUR 4.7–12 billion (each based on capital stock portions). This results in total damage of EUR 11 to 29 billion (cf. also Schäfer et al., 2021). However, it must be taken into account that the flooded areas represent only a part of the total affected area, so that the extrapolation of the damaged area was associated with high uncertainty at the time.

Shortly after a disaster occurs, there is usually considerable uncertainty about the level of damage, so that such simplified

methods are reasonable to provide timely loss estimations. In the case of the river Ahr, for example, this was necessary because no precise flood levels were initially available. Subsequently, these assessments were continuously improved–whenever new satellite products or analysis techniques became available for enhanced damage assessment.

As a result, since summer 2021, various loss assessments have been made by both policy makers and the insurance industry (insurance associations, reinsurers, catastrophe modeling company), selectively compiled in Table S2 in the Supplementary

material. At the end of 2021, GDV estimated insured losses for the July 2021 flood event at EUR 8.2 billion in Germany alone and EUR 11 billion for all affected countries (GDV, 2021b). However, only about 37 to 47 % of residential buildings were

---

[1]The GRADE approach, developed by the World Bank and supported by the Global Facility for Disaster Reduction and Recovery (GFDRR), is designed to provide governments and other key stakeholders involved in post-disaster damage assessment, relief, and recovery phases with initial rapid estimations of physical post-disaster damage within 2 weeks.





insured against floods, making the total damage significantly higher. In addition, infrastructure was massively affected (see also Sect. 4.3).

The German government estimated the total flood damage at EUR 32.05 billion, with 57 % of the damage attributable to
RP and 41.5 % to NRW; the remainder is attributable to damage in Bavaria and Saxony (based on the EU solidarity fund application of the German government within the final report from March 2022; BMI, 2022). These damage estimates include reconstruction costs compared to some of the damage-only estimates in Table S2. However, it must be taken into account that by the time of writing it is still unclear what the final reconstruction costs and the associated damage costs will actually be.

Most reinsurance companies currently use the GDV or German government estimations, both of which are closely aligned
(depending on the exact definitions). This would make this event one of the two largest events for natural hazards damage in Europe in the last 43 years (after the 1980 Irpinia earthquake in Italy) and definitely one of the five largest according to the European Environment Agency (based on CATDAT; EEA, 2022). The difference between book value damage, replacement costs, and reconstruction costs has yet to be determined and will likely be identified by follow-up analyses, meaning that the final economic damage value for comparison with other past disasters is not known at this point.

## 4.3 Affected rail and road infrastructure

The flood hit both the federal states of NRW and RP heavily concerning the transport and traffic sectors, both of which are classified as critical infrastructures. Infrastructures are considered 'critical' when they are of significant importance for the ability of modern societies to function (BMI, 2009). Therefore, disruption in rail and road infrastructure can have serious consequences. In the Ahr Valley alone, an estimated 103 bridges were damaged or completely destroyed (cf. BMI, 2022).

Most of the disruptions regarding rail and road occurred directly after the event. On 15 July 2021, 4 % of the total road traffic reports issued by the police in RP and NRW (Sect. 2.1.5) were directly related to the flood event. Given the large size of the two states and a generally high number of traffic reports and since this does not include indirect effects such as traffic jams, this can be considered a high percentage. In total, 39 road sections and 33 rail lines were affected. This decreased to 1.5 % on 23 July (15 rail lines, 15 road sections), but changed little until 31 July (11 rail lines, 11 road sections). At the end of August 2021,
1 % of total traffic reports issued in the two federal states was related to the flood event. At the end of September, this finally decreased to less than 0.3 %. The number of disruptions in rail traffic decreased only marginally until the end of September, while in road traffic there were only two reports related to the flood.

The fact that there were still significantly more rail than road disruptions remaining in fall suggests that repairing rail infrastructure is more time-consuming than repairing road infrastructure. According to a board member of the DB, the extent
to which rail infrastructure was hit by the flood is historically unique (SPIEGEL, 2021a). This is because the event was concentrated primarily on some regions, but was particularly severe in these. In NRW alone, around 600 km of railroad tracks were affected (Deutsche Bahn, 2021). When comparing traffic affected by the July 2021 flood with the situation of past flood events, for example the 2013 European floods (Thieken et al., 2016), the total number of disruptions for both rail and road traffic was much lower in 2021. This is likely due to the rather local scale of the flood event in 2021 as compared to 2013,
where the flood affected a major part of eastern Germany (e.g., Fig. 7 in Thieken et al., 2016).


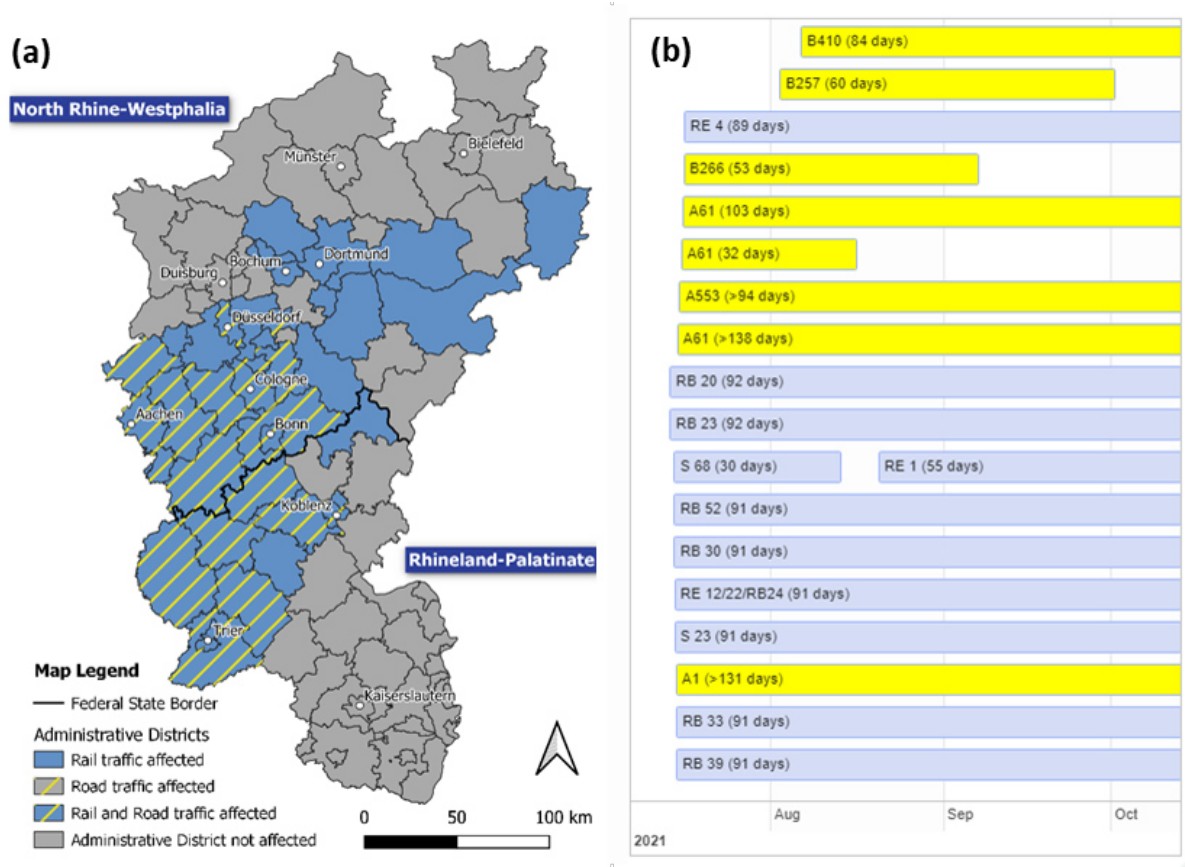

**Figure 11.** (a) Affected rail and road infrastructure as in different administrative districts in NRW and RP. If any rail track (i.e. *Regionalbahn*, RB; *Regional-Express*, RE; *S-Bahn*, S) or highway (i.e., *Autobahn*, A, or *Bundesstraße*, B) within the district is affected, the district is counted as affected; and (b) timeline of rail (blue) and large road (yellow) disruptions, regardless of the severity and including flood-induced construction work, with a duration of $> 25$ days. Roads or rail lines are listed separately if different time frames are involved.

Administrative districts along the federal state border and neighboring administrative districts of both federal states were most affected (see Fig. 11a). About twice as many districts were affected by rail disruptions as compared to road disruptions. One possible explanation is that disruptions in train connections usually affect a longer section of track, while disruptions on the road affect shorter segments. Figure 11b shows that long-term disruptions ($> 25$ days, equaling to 22 % of the total disruptions)
mainly affected rail infrastructure (blue) in comparison to road infrastructure (yellow; as of mid-October 2021). Regional trains (i.e. *Regionalbahn,* RB; *Regional-Express*, RE; *S-Bahn*, S) were especially affected by long-term damage, whereas on the road, especially the highways A1 and A61 were impacted: The foundation of a bridge close to the town of Hürth (south of Cologne) was washed away, which led to the sagging and rupture of the bridge. Furthermore, at the interchange of the town Erftstadt (Erft catchment), the river Erft destroyed a 100 m noise barrier causing two traffic lanes to break off.



In mid-October 2021, 30 km of highways were still completely blocked, which equals 23 % of the roads blocked directly after the flooding (Schmitz, 2021). Construction works regarding road infrastructure are expected to continue until summer 2022, which means that some sections of the road are not accessible until then (ADAC, 2021). Regarding rail infrastructure, experts struggle to give an estimation on how long the reconstruction will take. In mid-December, DB issued an update on the post-flood reconstruction work. Nevertheless, important connecting lines have already been opened, for example, on the Ahr

Valley railroad (Ahrtalbahn; SPIEGEL, 2021b).

    For transport infrastructure alone, the German federal government estimated a damage of EUR 2 billion (MDR, 2021). Damage regarding rail infrastructure alone is expected to amount to EUR 1.3 billion (BMDV, 2021). A task force has been set up in the Federal Ministry for Digital and Transport to carry out a further assessment of the damage. The task force's duties include to determine the expected costs and to press ahead with reconstruction (BMDV, 2022). Reconstruction shall happen as

soon as possible and is therefore focused on a 1:1 replacement in the current state of the art. However, the complete restoration of the heavily damaged rail lines will only be possible in the coming years, as this reconstruction partly means a entirely new construction (BMDV, 2021).

## 5   Discussion and conclusions

The July 2021 flood in western Germany and neighboring regions was one of the five most severe and expensive natural

catastrophes in Europe in the last half century. More than 180 people lost their lives, and well over 10 000 buildings were damaged. Numerous critical infrastructures, such as power and water supply networks, bridges, railway tracks, and roads, were destroyed partly or completely. In total, 39 road sections and 33 rail lines were damaged by the flood, which significantly hampered relief deliveries and clean-up work in the affected area. According to estimates from the German government in March 2022, the total damage in Germany alone amounts to EUR 32 billion (BMI, 2022), although it is still unclear how

high the final reconstruction costs will actually be. The overall scale of the disaster surprised both the public and scientific community.

    As part of CEDIM's Forensic Disaster Analysis (FDA) in near-real time, an interdisciplinary team of scientists investigated this severe flood event from different perspectives. Based upon analyses of available model and observational data and supplemented by own simulations and assessments, this paper examined the complex interactions among meteorological, hydro-

logical, hydraulic, and geomorphological processes and mechanisms that led to the extraordinary flood. Further, we attempted to estimate the impact in terms of traffic disruptions and economic losses, although the exact numbers still remain uncertain as reconstruction is ongoing. Some of the analyses presented were conducted while the flood was still ongoing–such as the estimation of inundation areas from photographs, video sequences, and satellite overflights, and derived damage assessments. Not only the flood's intensity was decisive in the outcomes, but also the failure of the warning chain (Fekete and Sandholz,

2021; Thieken et al., 2022). The population was informed too late or insufficiently about the extent of the imminent disaster despite the fact that both deterministic and ensemble weather forecasts issued, for example by DWD and ECMWF, predicted extreme rain totals and signified the potential the occurrence of a very unusual or extreme event at least 2 days before the





flood. However, because of the high political relevance, the failure of the warning chain was not discussed here. Parliamentary investigation committees have been installed in RP and NRW to investigate these failures and possible political consequences.

From a meteorological perspective, several factors combined led to the extremely high rain totals and severe flooding. First, during a period of approximately 3 weeks before the event, persistent atmospheric blocking with corresponding subsidence and cloud dissipation over the Baltic Sea resulted in a significant sea surface temperature (SST) anomaly (up to 6 K compared to the climatological mean). This allowed considerable amounts of water to evaporate, such that the air masses in the boundary layer over the Baltic Sea were exceptionally moist. According to our trajectory analysis, the easterly flow on the northern flank

of the low-pressure system *Bernd* transported these moist air masses to the affected region. The still prominent high pressure center over eastern Europe blocked the eastward displacement of the low-pressure system *Bernd*, which in the days before the flood moved very slowly with its frontal system from northern Italy over southern and northern Germany until it finally moved retrograde towards NRW and RP. This fact and the orientation of the main rainfall area nearly parallel to the propagation direction resulted in very high local rain totals, which affected a large area within approximately 15 hours. Locally embedded

convection further intensified the stratiform precipitation. According to extreme value statistics, the observed rain totals of up to 150 mm correspond to return periods of more than 500 years over a large part of the study area. Finally, two additional factors were crucial for the record flood. Because of wet conditions in June and July, the antecedent precipitation index showed an above average wet period with wet soils in the affected area. Combined with the steep slopes of the Ahr and other river valleys in the region, this supported rapid rainfall-runoff transformation by overland flow through infiltration and saturation excess.

Because of the long duration of heavy rainfall over an extended area, the water levels in the headwaters that originate in the central part of the Eifel plateau rose very rapidly to unprecedented heights. All peak values were clearly beyond the peak discharges of a 100-year flood event ($HQ_{100}$), at some gauges even by a factor of more than 5. At the downstream gauges, peak water levels were reached some hours later, in the early morning of 15 July, but with an even more dramatic increase. Estimated peak discharges were all clearly above the statistical $HQ_{100}$ values, with a record factor of around 7 at the gauge

Bliesheim/Erft. Comparably dramatic rises and peak discharges were also reported east of the Rhine (e.g., gauge Hückeswagen with a peak discharge nearly 3 times larger than the largest flood recorded previously and the $HQ_{100}$), and for several rivers in Belgium and Luxembourg.

The flood's large magnitude posed a great threat to many retention basins in the study area. For example, at retention basin Horchheim (river Erft), the peak inflow was estimated to be about 4 times higher than the statistical $HQ_{10\,000}$, which is the

design value for ensuring dam stability. Although all floodgates were opened, the dam still overflowed, causing severe damage to all service infrastructure and partial destruction of the dam by backward erosion. Similar situations were observed at several other retention basins.

Various hydro-morphodynamic processes that condition the valley response to the flood, as well as changes in the valley morphology during the event exacerbated the impact of the flood. As a detailed assessment of all hydro-morphodynamic effects

is very complex, we selected and discussed a few paradigmatic cases of fluvial morphology, sediment and debris flow, and landscape singularities to highlight the processes and effects involved. Local scour, bank erosion and collapse destroyed several gauging stations (e.g., Altenahr), which hampered hydrological monitoring of the event in real-time. Hillslope denudation and





widespread landslides yielded an excess of sediment in the river channels network, which in turn contributed to changes in morphology that interacted non-linearly with the flood propagation. Post-event images indicated that a large amount of debris was transported in the congested and hyper-congested regime, with severe consequences for flood propagation and effects. Flood levels increased in response to extra resistance to the flow and because of partial or complete blocking of cross sections available to the flow. Other effects relevant to the higher flood levels are anthropogenic morphology changes in the river cross section attributable to railroad bridges (increased bottleneck effect) and probable damming and clogging of the river cross sections at the bridges attributable to the trapping of debris transported from the upstream valley. Further, it can be assumed that a larger number of assets and industrial components (vehicles and caravans, bins and containers, construction materials) caused a higher volume of debris available for recruitment in the upstream valley. All these effects may explain why the level of the 2021 flood at several sites was higher (e.g., Dernau by 2.4 m) compared to the 1804 flood, although the estimated peak flood discharges were in the same order of magnitude.

Remote sensing of rapidly available imagery from social media, television, and news media proved to be crucial for estimating the inundation area during the first days of the disaster to estimate the socio-economic impact. Satellite imagery was initially only a supporting data set because of limited quality due to either cloud cover or ambiguous interpretation. However, the inundation area only captured a subset of the overall affected region, which not only suffered from direct losses due to damaged buildings and fatalities, but also through a variety of indirect losses due to interruptions of critical infrastructure and transport routes.

Based upon our analyses, we draw the following conclusions and make recommendations to improve the estimation of flood hazard and risk:

- Catastrophic events such as the July 2021 flood occur very rarely because, as described above, the relevant processes and mechanisms that amplify the event and act on different spatial and temporal scales must interact optimally. However, as such very rare events (referred to as gray or black Swans according to Taleb, 2010) are responsible for the greatest number of fatalities and highest economic losses, our still insufficient knowledge must be further improved through more dedicated research.

- The flood hazard (e.g., a 100-year flood level) typically used as a design basis for flood protection is currently underestimated when historical severe floods are not considered. In the case of the Ahr, historical analyses have shown that events comparable to the July 2021 flood had occurred already in 1804 and 1910, but only data from the continuous records (e.g. for gauge Altenahr starting in 1946) were used to estimate $HQ_{100}$ values. Furthermore, when analyzing and interpreting hydrological data, it is important to consider that infrastructures, landscape occupation, and flood protection measures often change considerably with direct consequences on flood risk over time scales shorter than the return period considered for flood emergency management.

- In recent years, the capability of weather models has improved significantly due to increasing computational power, better observations, advanced model physics and data assimilation, higher resolution, and larger ensembles (Bauer et al., 2015). While the meteorological potential for a extraordinary event in the region was identified in the DWD and ECMWF





forecasts at least two days in advance, it remains difficult to communicate the probabilistic nature of forecasts, in particular with respect to extremes, to users and the general public. Here, we have used the EFI based on ECMWF forecasts as a simple and intuitive metric that successfully gave indications for a potentially harmful event already several days before
it occurred. We advertise usage of such concepts for other cases and forecasting systems to facilitate communication. However, a robust assessment of the model climate and its extremes as a reference is necessary to compute the EFI, which requires expensive re-forecasts. Currently, ECMWF is one of few operational centers internationally that provides such a metric operationally.

– By interacting with the landscape and network morphology of river channels, flowing water enhances a flood's de-
structive power. Relevant effects include: (a) the occurrence of extreme landscape erosion, including rapidly developing rivulets and landslides; (b) rapidly evolving erosion and scour processes in the channel network and urban space; (c) recruitment of large-scale debris from the natural landscape and urban landscape; (d) deposition and clogging of bottlenecks in the channel network with eventual collapse; (e) possible intersection of the flow with landscape anthropogenic or natural singularities; (f) interaction with vegetation. None of these aspects are considered in the current practice of
flood hazard modeling, which leads to an underestimation of the actual risk (cf. also Dietze et al., 2022). We suggest an update of the Floods Directive, which should require that flood hazard and risk assessments include a heuristic hydro-morphodynamic approach that considers the landscape and network of river channels, including sediment transport and morphology changes (Nones, 2019).

– River gauges are a crucial source of information in the event of a flood. These gauges should be installed in such a
way that they also function reliably in the event of an extreme flood. Therefore, particular care should be taken when replacing the river gauges lost during the recent flood to avoid similar issues in the future. This is also true for other river basins with similar characteristics.

– CEDIM's FDA task force (Schäfer et al., 2021) has demonstrated that inundation areas and damage can be estimated reliably from photos and video sequences in near-real time during or after an event–ideally within a few hours or days.
Such results, made available to authorities or relevant stakeholders, can help to get a quick and better overview of the overall situation in order to assess the damage more quickly and to respond more appropriately in the aftermath of catastrophic events. This helps to mitigate associated adverse effects

– The impacts of extreme hydro-meteorological events also depend upon the way both authorities and individuals respond to predicted extremes and whether they take appropriate action (e.g., securing valuable assets and critical infrastructure,
evacuations). Operational hydrological forecasts in Germany, however, currently do not provide impact-based information about an upcoming event, but only about discharge and water levels at the respective gauges. To be as prepared as possible for upcoming extreme events, accurate predictions of both their physical characteristics and expected impacts on society and the built and natural environment are essential (Taylor et al., 2018; Merz et al., 2020; WMO, 2020). Impact-based forecasting that incorporate exposure, vulnerability, and social systems in addition to the hazard have great


potential to reduce damage and increase resilience substantially. The feasibility of this was recently been demonstrated by Apel et al. (2022), who used a simplified hydrodynamic flood model to retroactively incorporate spatially explicit information (such as inundation area, depths, and flow velocities) based on predicted gauge discharges or water levels into current hydrological forecasting systems for the July 2021 flood. This provides real-time information on the expected extent of flooding and its impacts. By implementing the model on graphical processing units, simulation times are within
a range required for operational flood warning.

PART2 of the paper (Ludwig et al., in prep.) puts the July 2021 flood in historical context. This is accomplished not only by statistical analysis of observational data, but also by incorporating a large ensemble of regional climate simulations (comprising 12 000 years; cf. Ehmele et al., 2020, 2022). The second focus of PART2 is to examine how the precipitation causing the flood event could unfold in the context of climate change. The related analysis is based on a storyline approach using a series of
pseudo global warming experiments and on a conventional ensemble of future climate projections.

*Code and data availability.* HYRAS-DE, RADOLAN, KOSTRA, and German precipitation station data, all from DWD, are freely available for research at the Open Data Portal (https://opendata.dwd.de, last access: 9 May 2022). HYRAS data can be requested at DWD for research and education purposes. DWD weather forecasts (ICON-EU, ICON-D2, and ICON-D2-EPS) are available at Pamore (PArallel MOdel data REtrieve from Oracle databases) after registration (https://www.dwd.de/EN/ourservices/pamore/pamore.html, last access: 9 May
2022). The ECMWF-EPS data are available at the ECMWF Meteorological Archival and Retrieval System (MARS) after registration (https://apps.ecmwf.int/archive-catalogue/, last access: 9 May 2022). ERA5 data are freely available at https://apps.ecmwf.int/data-catalogues/era5/?class=ea (last access: 9 May 2022). The LAGRANTO documentation and information on how to access the source code are provided in Sprenger and Wernli (2015). The GEM data are available at https://dd.weather.gc.ca/model_gem_global/15km/grib2/lat_lon (last access: 9 May 2022). River gauge data are available on request from the responsible water authority: Water administration of Rhineland-Palatinate
(https://www.lfu.rlp.de, last access: 9 May 2022) for gauges Müsch, Altenahr, Jünkerath, Kordel, Prüm 2, and Prümzurlay; Erftverband (https://www.erftverband.de, last access: 9 May 2022) for gauges Schönau and Bliesheim; Wupperverband (https://www.wupperverband.de, last access: 9 May 2022) for gauges Hückeswagen and Opladen. Sentinel-1 and Sentinel-2 data can be freely accessed at https://www.sentinel-hub.com (last access: 9 May 2022). Information and data regarding traffic are available on request to Deutsche Bahn Press Office and on www.Stau1.de. CATDAT data are available at https://www.eea.europa.eu/data-and-maps/data-providers-and-partners/risklayer (last
access: 9 May 2022).

*Author contributions.* All KIT authors jointly designed the research questions of the study, continuously discussed the results, and wrote the text passages for their respective contribution. SM coordinated the joint collaboration, wrote abstract and introduction, and prepared the final version of the paper. FE, BM, JQ, and PL were responsible for the meteorological analyses with the help of ACA, HF, MH, and JGP (including description of data and methods). UE, along with CG and MS, was responsible for the hydrological analyses (including
description of data), with the latter two providing additional river gauge data. MF and FS were responsible for the analysis and discussion of hydro-morphodynamic processes. AS was responsible for remote sensing analysis and inundation area modeling, and JD was responsible



for damage assessment, including further damage classification. KK and CW analyzed the affected rail and road infrastructure. MK wrote the discussion and conclusion. Manuscript revision and editing mainly involved SM, UE, MK, PL with help from FE, PK, and JGP.

*Competing interests.* One of the coauthors (JGP) is a member of the editorial board of *Natural Hazards and Earth System Sciences*. The

peer-review process was guided by an independent editor, and the authors have also no other competing interests to declare.

*Acknowledgements.* This study is the result of an interdisciplinary collaboration at KIT, originating from the CEDIM's Forensic Disaster Analyses (FDA) on the flood of July 2021 in summer 2021. The Center for Disaster Management and Risk Reduction Technology (CEDIM) is a cross-disciplinary research center in the field of disasters, risks, and security at the Karlsruhe Institute of Technology (KIT) funded by the KIT and the research program "Changing Earth – Sustaining our Future" in the Helmholtz research field "Earth and Environ-

ment". Several authors acknowledge partial funding from BMBF "ClimXtreme Module A" (01LP1901A), BMBF "RegIKlim-NUKLEUS" (01LR2002B), BMBF "RegIKlim-ISAP" (01LR2007B) and DFG "Waves to Weather" TRR 165. Additionally, PL has been supported by the Helmholtz Association (Climate Initiative REKLIM grant) and JQ's contribution was funded by the Young Investigator Group "Sub-seasonal Predictability: Understanding the Role of Diabatic Outflow" (SPREADOUT, grant VH-NG-1243). JGP thanks the AXA Research Fund for support (https://axa-research.org/en/project/joaquim-pinto, last access: 9 May 2022). The authors thank the DWD, ECMWF, CMC,

the Copernicus program and the responsible water authority (Water administration of Rhineland-Palatinate, Erftverband, Wupperverband) for providing different observational data. UE thanks M. Göller and N. Demuth from the water administration of Rhineland-Palatinate and N. Patz from water administration of Luxembourg for providing valuable support related to gauge data and reconstruction of the flood event. KK and CW thank the Deutsche Bahn Working Group IT Platform and Services Traveler Information and Stau1.de for the provision of the traffic data. SM and BM thank Robinson et al. (2014) for providing the EarthEnv-DEM90 digital elevation model data set and Natural Earth

https://www.naturalearthdata.com (last access: 11 May 2022). We thank all the private contributors of imagery during and after the flooding, which helped us identifying the inundation area. Many thanks also to the photo- and videographers (Heinz Grates, Martin Seifert, Bettina Vier, Maurice), who allowed us to use their photos or screenshots of their footage within this publication. Finally, we thank the open-access publishing fund of KIT.





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
