# Peer review of "A multi-disciplinary analysis of the exceptional flood event of July 2021 in central Europe. Part 1: Event description and analysis"

_Natural Hazards and Earth System Sciences, 2022_

## Referee Comment (RC1)

**General comments**

In their manuscript *"A multi-disciplinary analysis of the exceptional flood event of July 2021 in central Europe. Part 1: Event description and analysis"*, Susanna Mohr and colleagues provide an overview of the disastrous July 2021 flood event.

The authors have done an impressive job in collecting and compiling information on various aspects of the July 2021 flood, and the sheer effort behind this needs to be appreciated.

Having said that, the objectives and specific research questions related to this study remain unclear to me. In ll. 74 ff. of the introduction, the authors state that *"[...] the objective of this two-part study is a multi-disciplinary assessment of the entire process chain of the July 2021 flood in central Europe - from causes to impacts to historical classification and climatological context [...] While Part 1 focuses on the description of the event across various disciplines (meteorological, hydrological, hydro-morphological, economic) [...]"*. This is not a research question, and nowhere in the paper, the authors specified their idea of a *"multi-disciplinary assessment"*. In ll. 754-755 of the conclusions, they state that *"this paper examined the complex interactions among meteorological, hydrological, hydraulic, and geomorphological processes and mechanisms that led to the extraordinary flood [...]"*. I find this statement difficult to confirm: instead, the manuscript largely remains a *description* from different sub-disciplines (meteorology, hydrology, hydro-geomorphology, impacts/damages), listed one after another, but mostly not related to each other in terms of an *analysis*. This is a challenge that many papers have to cope with when they aim to provide a holistic assessment of an extreme event - I noticed a similar referee comment on the paper of Caldas-Alvarez et al. (2022) about the Berlin 2017 event in this same special issue (link).

While it is, in the direct aftermath of such a disaster, valid and required to focus on the rapid compilation of information and data, and to make such compilations available to decision makers, the research community, and the general public, I wonder whether we should have, by now, reached a phase in which the research community should find more well-defined modes of event analysis. Then again, I am aware that putting together such a manuscript requires a lot of time, and that the processes for this probably had already started in 2021.

My overall recommendation is that the authors take a step back and re-evaluate the purpose of this paper. What is your scientific objective, aside from compiling as much as we know about the event? I think it would help the paper very much to, after a brief synopsis of the event, identify maybe two or three important and *specific* research questions (e.g. with regard to specific interactions), and tell the story along these questions - instead of just listing one discipline after the other. Many aspects of the paper are quite interesting in itself, but maybe not required for a holistic view? Do they have to be a part of this study? For example, I found it difficult to understand why we need the section on the "synoptic overview and atmospheric characteristics". And the predictability of heavy rainfall (weather forecast analysis) would be relevant only if the authors had actually investigated the failure of the early warning chain and the resulting implications on the impacts in terms of damage and loss of lives. Also the interaction between discharge dynamics, hydro-morphodynamics and impacts is not yet sufficiently elaborated.

I understand that it is challenging to revise the paper along these lines, but I am confident that the authors will find an adequate way. Is it helpful to recommend that the paper should be much shorter? It almost took me three days just to work my way through it. I think the length of the paper could be easily reduced by at least a third and I hope this will help separating what is relevant from what is not.

One more comment on the issue of multi-part papers: Surely it is up to the editorial team to assess whether a multi-part publication is warranted. Personally, I have never really understood the need for multi-part papers. A paper should be self-sustained, and *evaluated* as an individual piece of scientific work. This is also the basis for this review. Of course, papers can and should refer to each other and build upon each other, but in my view, that does not require an explicit multi-part approach. In the present context, the multi-part approach supports the impression that PART1 is more about compiling "everything we know" instead of asking and addressing well-defined research questions.

**Specific comments**

**Is this a research article or a review?**

To be honest, I am also not sure which type of manuscript I am dealing with. Over large parts, the manuscript more resembles a review paper instead of a research article. It struck me that there is no formal "results" section (instead "event description and analysis"). This is not a fundamental issue in itself; however, it again points to the fact that there are no specific research questions and hence no specific results to address these. Furthermore, the content in section 3 ("event description and analysis") is largely not based on the data and methods which have been described in section 2 ("data and methods").

I would like to discuss this in detail: Section 2.1 (data) documents precipitation data, atmospheric model data, river gauge observations (not the data/methods on reconstruction of water levels/discharge), Sentinel1/2 data (for inundation mapping), and traffic data (reports on road and railway disruptions). Section 2.2 (methods) documents the trajectory analysis (for moisture source analysis), extreme value statistics for precipitation, and the computation of the antecedent moisture index. Together, section 2 incompletely addresses the methods and data that were used to put together section 3; in section 3.2, for instance, reconstructed water levels/discharges play an important large role; section 3.3 (hydro-morphodynamic processes) is almost entirely unrelated to data and methods documented in section 2; and section 4 (impacts and consequences), too, is based on methods and data sets (e.g. aerial/media footage, loss models, insurance data) which were not mentioned in section 2 (except the Sentinel data and the reports on traffic disruptions).

Altogether, I understand that the variety and mass of methods and data that section 3 is based upon is almost impossible to describe in section 2. The reason for that is that much of the content shown in section 3 is not really based on the application of data and methods in the context of study, but rather a compilation and synthesis from other recent studies about the July 2021 event, namely Fekete and Sandholz (2021), Dietze et al. (2022), Thieken et al. (2022), Schäfer et al. (2021), Apel et al. (2022), BM (2022), Roggenkamp and Hergert (2022).

Hence, I would like to ask the authors to clarify, from the beginning, how they combine the original analysis of data and methods with the results of other studies in the context of the paper. Furthermore, I suggest that the authors publish the data which they used for this study in a single dedicated and documented data set to accompany this manuscript as an asset, even if parts of this data are available elsewhere (like the precipitation data or the Sentinel data). That way, they would not only make the original contribution of this study more transparent, but it would also be a valuable service to the research community. I understand that this will require some coordination with the source institutions, e.g. for the river gauge observations or the railway disruption data. Still, I think it'll be worth the effort.

**Text is reproducing figures and tables**

Very often, the content of figures and tables is reproduced/reiterated in the main text. I think the authors should trust more in the information content of their figures, and if they don't, figures should be made more concise.

**Introduction and conclusions sections do not sufficiently frame the study**

The introduction reflects the general issue of this paper: instead of using the introduction to systematically develop and justify specific research questions, it appears to string together previous studies in a rather unrelated fashion.

Similarly, the section "discussion and conclusions" (four pages!) provides a long list of statements, and for many of them, it is not really clear how they are based on the results presented in this study and how they relate to a study objective.

**Other comments**

**Fig. 1:** Please make sure that the border of Luxembourg is visible behind the river Sauer; I am not sure why the catchments are hatched - it does not improve the readability of the map. Furthermore, please show the catchment boundaries for all river catchments discussed in this paper (see section 3.2).

**L. 90:** While we see an area in the map, it remains unclear what signifies the "study area". Which parts of the area are actually *studied*? Is it the area in which specific precipitation totals were exceeded? Is it a combination of catchments, and if yes, which?

**Ll. 93-100:** I am not sure how helpful this paragraph is for the audience. If specific districts or municipalities are important in a spatial context, they should be included in a map. If the main map is too crowded or its scale to small, you can provide another inset or sub-plot which e.g. focuses on the administrative structure within e.g. the Ahr and the Erft catchments.

**Ll. 125-130:** Why is a weather forecast model used to analyse geopotential patterns and precipitable water? Why not use an analysis? The section header says "Weather forecast and analysis data", but the section does not describe any analysis data.

**Ll. 146:** Again, you mention the study area with regard to the selection of river gauges. Please explicitly specify, e.g. in Fig. 1, what you consider as the study area. You also specify

a lot of selection criteria; but why did you not just use, in the context of your study, all river gauge data that were available to you?

**Ll. 153 ff.:** I think the reconstruction of water level or discharge is an important methodological feature and should hence be described or adequately referenced.

**Ll. 180 ff.:** You are using ERA5, but you did not specify ERA5 as data used in this study in section 2.1.

**L. 195:** you say *"for example, [...] daily precipitation totals"*. Which variables, apart from that, were subject to extreme value statistics?

**L. 239:** *"very high values of total precipitable water of more than 40 kg m$^{-2}$ were reached, which occur only very rarely."* - how rarely?

**Ll. 256 ff.:** The authors end the section with the sentence *"Heavy precipitation associated with quasi-stationary low pressure systems, their fronts, or convective systems located on the western flank of persistent blocking systems is common in Europe during summertime, so the large-scale situation is not unusual"*. So what do we actually learn from the section *"Synoptic overview and atmospheric characteristics"*? How does it help us to understand how the event unfolded?

**Ll. 262:** How do you know the presence of embedded convection?

**L. 269:** please quantify what you mean by *"major part"*.

**Fig. 3:** Why not zoom all maps into the study area? I do not find it helpful to show all of Germany. For the API, it would also help to show return periods in order to appreciate whether/how the API was anywhere near unusual for this event.

**Fig. 4:** for the sake of comparability, please use the same x-axis for subplots a and b; I assume the black dashed line (observed reference from RADOLAN) is the spatial average - but over which region? Please specify in the figure caption.

**Ll. 319 ff.:** At some point, you need to provide a bit of context on the interpretation of the EFI - also quantitatively: How frequent is an EFI exceedance > 0.8 in the region, hence how well does it signify the potential for such a singular event?

**Ll. 343:** *"twice the climatological mean"* is not very helpful, in my opinion, as the reader is not informed about the statistical distribution of API values in the area. It would be more informative to provide a (extreme value) statistical assessment in terms of frequency.

**Ll. 345 ff.:** *"In the southern parts of the Eifel, the Ardennes in the north-west, and in the north-east of the study area in the Wupper region, generally less than 10 mm of soil water storage were still available for infiltration. In the remaining regions, free soil water storage was larger, but still below average, ranging mainly between 10 and 30, sometimes 75 mm."* On which basis are these statements made? The API does not allow for such an assessment (by the way, since the API is only computed from precipitation, it is basically a meteorological quantity, not a hydrological one). More importantly, in the context of your study and its stated objective, it would be crucial to assess the extent to which drier soils

(e.g. average soil moisture in July) could have been able to retain substantial amounts of water and hence significantly reduce the hydrological response.

**Ll. 368 ff.:** Apart from the fact that peak flow was exceptionally high with a steep rise, I am not really sure what to learn from the entire section about "Ahr, Kyll, and Prüm river basins". I think it would be a perfect opportunity to analyse why quite a similar rainfall in the headwaters lead to different event amplitudes in the different basins. These lines (ll. 368 ff.) aim at that, but I have the impression that the statements are rather based on speculation.

**Fig. 5 and ll. 373 ff.:** How were the time series reconstructed (dashed lines)? I understand that peak water level and hence peak flow can be reconstructed from debris lines etc., but how about the recession? How can we explain the high water level at gauge Schönau after the peak, while discharge had already recessed to normal levels?

**Ll. 456 ff. / section 3.3:** many statements in section 3.3 are not identifiably based on scientific data - either observation or models -, but seem to be speculative, hear-say, or partly reproducing textbook knowledge. The actual scientific contribution of this section remains unclear to me, especially in comparison to the paper of Dietze et al. (2022) which is widely cited by the present study.

**Fig. 9:** I think that the scale of this figure is not really helpful for appreciating the inundated area.

**Ll. 589 / section 4.1:** I think the issue of rapid inundation mapping is an important research topic, but I do not see how it is relevant in the post-event analysis almost one year after the fact.

**Ll. 630 ff. / section 4.2:** The same applies to section 4.2: Why is the rapid damage estimation process important? In the context of this study, shouldn't the best possible estimates of damages and damage processes be used? Furthermore, section 3.3 emphasised very much the role of debris/sediment/hydro-morphodynamics for damage processes, but I do not find this issue in section 4.2. Isn't this damage process chain one of the key properties of the July 2021 event?

**Ll. 700 ff. / section 4.3:** This section is mainly based on the number of traffic reports and the number of affected railroad sections, irrespective of the severity of disruption. Doesn't that limit the meaningfulness and the comparability to the 2013 event, given that it should matter whether a road/bridge is just disrupted or whether it actually disappeared? Maybe the persistence of the disruption for specific sections/lines could be interpreted as a proxy for severity, but there should be a reference from other events to compare to.

**Fig. 11b:** In total, I find subplot b not very informative, particularly since most of the specific road and railway numbers are not referenced in the main text. Wouldn't a simple plot of affected sections/lines over time be more informative and concise?

**Ll. 744 ff.:** *"The July 2021 flood in western Germany and neighbouring regions was one of the five most severe and expensive natural catastrophes in Europe in the last half century"* - what is the source of that statement? How important is the consideration of EFI in the context of your study, given the apparently high skill of the ICON models?

**L. 763:** *"However, because of the high political relevance, the failure of the warning chain was not discussed here."* I understand, but why bring it up then? Could in this case the topic of predictability also be dropped from the manuscript?

**L. 809:** *"Remote sensing of rapidly available imagery from social media, television, and news media [...]"* - I don't understand what that means.

**Ll. 817 ff.:** *"[...] our still insufficient knowledge must be further improved through more dedicated research"* - is it really necessary to spell out the obvious?.

**Ll. 823 ff.:** Is this item a new conclusion drawn from this study? Wasn't this already pointed out by Roggenkamp and Herget (2022)?

**Ll. 849 ff.:** Sure, but how??

**Ll. 853 ff:** How is this a conclusion from the present study?

**Ll. 876 ff.:** Please see my above comment: I think it would be really helpful to publish the data along with this manuscript.

**Technical comments**

**L. 160:** Please replace "flood plain" by "inundated areas"

**L. 189:** were used

**L. 191:** 95th percentile

**L. 210:** suitability instead usability

**L. 224:** second "decade"?

**Ll. 272 ff.:** *"The long-term average for the month of July at this station is 69 mm (1981-2010); thus, in just a few hours, the rainfall added up to more than twice the usual monthly precipitation."* - This kind of statement is unnecessary.

**L. 307:** as early as

**L. 355:** replace same by similar

**L. 580:** replace "the hardest affected location" by "the location affected most severely"

**Fig. 10:** It impossible to distinguish EMSR517_v1 from EMSR517_v3

**Fig. 11:** the resolution is very low.

**L. 762:** delete "the occurrence"

**L. 831:** an extraordinary

**L. 837:** the term "re-forecast" sounds weird - isn't "hindcast" the correct term?

---

## Referee Comment (RC2)

**General comments**

The paper "A multi-disciplinary analysis of the exceptional flood event of July 2021 in central Europe. Part 1: Event description and analysis" by Mohr et al. gives an overview of the flood event last year, with a special focus on hydrological, and hydro-morphological processes and mechanisms. The paper describes very well the event across various physical disciplines. The complex interaction between those could be analyzed in more detail. The aspects of social science regarding the flood event are not addressed, at least the paper should underline of refer to the high importance of risk culture (e.g. risk awareness, risk communication).

**Specific comments**

L 1; L 16; L 69; L334, L744: Please think about if you want to use the term "natural disaster". There is no disaster without human interference, so it's never something "natural" Have a look at #nonaturaldisaster: https://www.nonaturaldisasters.com/

L 23: Figure 1 is mentioned here for the first time, but Figure 1 is currently in L 116. Why so far away?

L 35: displaced people?

L 39: in the meantime, flood hazard maps are updated see Roggenkamp & Herget. I would rather write the existing maps before and during the flood

L 107: see below – could you describe below more specific please?

**Technical corrections**

L 38: only one week

L63: six month

L73; L74: (e.g. ….)

L74: erosion, and

L105: one hour, but can reach up to one minute -> Numbers from one to twelve are written out

L128: used by

L129: In its global uniform resolution configuration it is run twice daily -> check the grammar

L307: as early as

L310: were predicted more than two days

L320: two days

L339: - namely soil wetness - -> missing spaces

L344: three weeks

L371: erosion, and

L372; L389f; L401, L484: - …. - -> missing spaces

L408f: the peak flow

L523: floods are?

L632: infrastructural, and

L762: two days

L857: This helps to mitigate associated adverse effects -> missing . in the end

---

## Author Comment (AC1)

**Reviewer #1**

**General comments**

In their manuscript *"A multi-disciplinary analysis of the exceptional flood event of July 2021 in central Europe. Part 1: Event description and analysis"*, Susanna Mohr and colleagues provide an overview of the disastrous July 2021 flood event.

We thank the reviewer for his/her time, the very careful and thorough review, and the many suggestions. We are aware that he/she took a lot of time.

The authors have done an impressive job in collecting and compiling information on various aspects of the July 2021 flood, and the sheer effort behind this needs to be appreciated.

Having said that, the objectives and specific research questions related to this study remain unclear to me. In ll. 74 ff. of the introduction, the authors state that *"[...] the objective of this two-part study is a multi-disciplinary assessment of the entire process chain of the July 2021 flood in central Europe - from causes to impacts to historical classification and climatological context [...] While Part 1 focuses on the description of the event across various disciplines (meteorological, hydrological, hydro-morphological, economic) [...]"*. This is not a research question, and nowhere in the paper, the authors specified their idea of a *"multi-disciplinary assessment"*. In ll. 754-755 of the conclusions, they state that *"this paper examined the complex interactions among meteorological, hydrological, hydraulic, and geomorphological processes and mechanisms that led to the extraordinary flood [...]"*. I find this statement difficult to confirm: instead, the manuscript largely remains a *description* from different sub-disciplines (meteorology, hydrology, hydro-geomorphology, impacts/damages), listed one after another, but mostly not related to each other in terms of an *analysis*. This is a challenge that many papers have to cope with when they aim to provide a holistic assessment of an extreme event - I noticed a similar referee comment on the paper of Caldas-Alvarez et al. (2022) about the Berlin 2017 event in this same special issue (link).

While it is, in the direct aftermath of such a disaster, valid and required to focus on the rapid compilation of information and data, and to make such compilations available to decision makers, the research community, and the general public, I wonder whether we should have, by now, reached a phase in which the research community should find more well-defined modes of event analysis. Then again, I am aware that putting together such a manuscript requires a lot of time, and that the processes for this probably had already started in 2021. My overall recommendation is that the authors take a step back and re-evaluate the purpose of this paper. What is your scientific objective, aside from compiling as much as we know about the event? I think it would help the paper very much to, after a brief synopsis of the event, identify maybe two or three important and *specific* research questions (e.g. with regard to specific interactions), and tell the story along these questions - instead of just listing one discipline after the other. Many aspects of the paper are quite interesting in itself, but maybe not required for a holistic view? Do they have to be a part of this study?

We will revise the content and structure of the paper. In particular, this includes that we will define new research questions to better highlight our findings and link them along an appropriate storyline. The two new research questions (RQs) are (roughly):

1) What were the hydro-meteorological causes of the July 2021 flood and what interactions / effects were observed? What made the flood so exceptional?
2) What can be presented shortly after an extreme event and how good are these first estimates? (context early response)

Another important change will be that we will introduce a new section that will be a "synopsis" and thus better bring together the results of the different disciplines (to counter the criticism that the results of the different disciplines are just "attached" to each other). In addition, this section will also be devoted to RQ2. We plan to create a new figure that will include a kind of timeline of what happened (and the interlocking processes) and the subsequent (possible) analyses. With this, we want to discuss the question of what can be conveyed immediately after an event, or what is possible in the context of a rapid event analysis/damage assessments; how good these can be and where the limits are (warning plays an important role here as well). Our objective with the section is to better highlight the "multi-disciplinary assessment" and the importance or capabilities of rapid forensic disaster analysis.

Regarding the " test reducing" aspect: Based on our new RQs, we think we are doing justice to all previous findings (from the first version) and that these should still be part of the study; however, we are aware that we should reduce within the sections in order to decrease the overall size of the paper. Also considering that a new section will be added (even if it will contain parts of the previous text).

For example, I found it difficult to understand why we need the section on the "synoptic overview and atmospheric characteristics". And the predictability of heavy rainfall (weather forecast analysis) would be relevant only if the authors had actually investigated the failure of the early warning chain and the resulting implications on the impacts in terms of damage and loss of lives. Also the interaction between discharge dynamics, hydro-morphodynamics and impacts is not yet sufficiently elaborated.

In terms of (extreme) weather events, it is always important to consider the large-scale context or processes and mechanisms, and to mention factors that contributed significantly (e.g., air mass transport, large-scale lifting processes, blocking). For example, atmospheric blocking was crucial that the Baltic Sea could be warmed so significantly and served as a source for the air masses. But we recognize that the paragraph also has the potential to be reduced.

Forecasting is an important part of the temporal progression (see timeline in the new section; comment above); furthermore, it is important for us to emphasize that the EFI is a simple and intuitive metric and can be used successfully to indicate a potentially harmful event several days before it occurs.

Regarding the last point: We will make this clearer in the new version.

I understand that it is challenging to revise the paper along these lines, but I am confident that the authors will find an adequate way. Is it helpful to recommend that the paper should be much shorter? It almost took me three days just to work my way through it. I think the length of the paper could be easily reduced by at least a third and I hope this will help separating what is relevant from what is not.
See answer above.

One more comment on the issue of multi-part papers: Surely it is up to the editorial team to assess whether a multi-part publication is warranted. Personally, I have never really understood the need for multi-part papers. A paper should be self-sustained, and *evaluated* as an individual piece of scientific work. This is also the basis for this review. Of course, papers can and should refer to each other and build upon each other, but in my view, that does not require an explicit multi-part approach. In the present context, the multi-part approach supports the impression that PART1 is more about compiling "everything we know" instead of asking and addressing well-defined research questions.
PART1 is an important basis for the second part; we hope this will become clearer when we submit PART2 in September.

**Specific comments**

**Is this a research article or a review?**

To be honest, I am also not sure which type of manuscript I am dealing with. Over large parts, the manuscript more resembles a review paper instead of a research article. It struck me that there is no formal "results" section (instead "event description and analysis").

This is not a fundamental issue in itself; however, it again points to the fact that there are no specific research questions and hence no specific results to address these.

Section 3 and Section 4 are our results sections. Personally, I find a heading that only contains the word "results" not very informative and prefer headings that make the content of the sections thematically clearer. Moreover, even an event description is an own work with a result, since the "important" processes/mechanisms are identified and discussed.

We suspect that our introduction initially gives the impression of a review paper, as our motivation was originally to refer to previous activities/studies in general on the flood event. In the revised version, we will focus only on studies that are relevant to our work or discuss similar aspects (current state of current research). We hope that the reformulating of the RQs will also make it clearer that our study is a "research article."

Furthermore, the content in section 3 ("event description and analysis") is largely not based on the data and methods which have been described in section 2 ("data and methods").

I would like to discuss this in detail: Section 2.1 (data) documents precipitation data, atmospheric model data, river gauge observations (not the data/methods on reconstruction of water levels/discharge), Sentinel1/2 data (for inundation mapping), and traffic data (reports on road and railway disruptions). Section 2.2 (methods) documents the trajectory analysis (for moisture source analysis), extreme value statistics for precipitation, and the computation of the antecedent moisture index. Together, section 2 incompletely addresses the methods and data that were used to put together section 3; in section 3.2, for instance, reconstructed water levels/discharges play an important large role; section 3.3 (hydro-morphodynamic processes) is almost entirely unrelated to data and methods documented in section 2; and section 4 (impacts and consequences), too, is based on methods and data sets (e.g. aerial/media footage, loss models, insurance data) which were not mentioned in section 2 (except the Sentinel data and the reports on traffic disruptions).

A large part of the data and methods is mentioned in Section 2, but we agree with the reviewer that this does not apply to all of it; some methods were not described sufficiently (e.g., level/discharge constructions) or were described in the results section (e.g., loss model). We will correct this in the revised version; this will also allow us to focus on the essentials in the results sections.

Altogether, I understand that the variety and mass of methods and data that section 3 is based upon is almost impossible to describe in section 2. The reason for that is that much of the content shown in section 3 is not really based on the application of data and methods in the context of study, but rather a compilation and synthesis from other recent studies about the July 2021 event, namely Fekete and Sandholz (2021), Dietze et al. (2022), Thieken et al. (2022), Schäfer et al. (2021), Apel et al. (2022), BM (2022), Roggenkamp and Hergert (2022).

Please note that Schäfer et al. (2021) is our own work and is the basis for this publication (however, it is only a technical report that was not reviewed). We also plan to move comparisons with other studies to the new synopsis section and discuss there, so that the separation to our own works becomes clearer.

Hence, I would like to ask the authors to clarify, from the beginning, how they combine the original analysis of data and methods with the results of other studies in the context of the paper. Furthermore, I suggest that the authors publish the data which they used for this study in a single dedicated and documented data set to accompany this manuscript as an asset, even if parts of this data are available elsewhere (like the precipitation data or the Sentinel data). That way, they would not only make the original contribution of this study more transparent, but it would also be a valuable service to the research community. I understand that this will require some coordination with the source institutions, e.g. for the river gauge observations or the railway disruption data. Still, I think it'll be worth the effort.

We find this a good suggestion; however, we must first clarify whether and in what form it is possible to integrate data from third parties.

**Text is reproducing figures and tables**

Very often, the content of figures and tables is reproduced/reiterated in the main text. I think the authors should trust more in the information content of their figures, and if they don't, figures should be made more concise.

With respect to Sect. 3.2 (Hydrological aspects), we would like to maintain that the joint discussion of the flood evolution over time in the text, and comparing them to historical and statistical peak flows, is a useful addition to the hydrographs as shown in Fig. 5 and listed in Table 1. Neither does the figure alone convey the points discussed in the text, nor can the text stand alone, without the figure. We therefore prefer keeping the text, in combination with the figure as is.

**Introduction and conclusions sections do not sufficiently frame the study**

The introduction reflects the general issue of this paper: instead of using the introduction to systematically develop and justify specific research questions, it appears to string together previous studies in a rather unrelated fashion.

Similarly, the section "discussion and conclusions" (four pages!) provides a long list of statements, and for many of them, it is not really clear how they are based on the results presented in this study and how they relate to a study objective.

Regarding Introduction: See answer above (less summary of other work related to the flood; more focus in context to our work; new research questions).

Regarding Conclusion: discussion parts from the Conclusion will become part of the new synopsis section, so we can focus more on a summary and outlook here.

Please also keep in mind that a large part of the section is also an outlook; the event will keep the research community (and decision makers) busy for some years and we have the opportunity to provide a food for thought for future activities.

**Other comments**

**Fig. 1:** Please make sure that the border of Luxembourg is visible behind the river Sauer; I am not sure why the catchments are hatched - it does not improve the readability of the map. Furthermore, please show the catchment boundaries for all river catchments discussed in this paper (see section 3.2).
We will redesign the readability of the map. To show all catchment areas would make the map confusing and go beyond the context of the paper; therefore, we will limit the revised version to our primary focus area and show only the catchment of the Ahr (without hatching).

**L. 90:** While we see an area in the map, it remains unclear what signifies the "study area". Which parts of the area are actually *studied*? Is it the area in which specific precipitation totals were exceeded? Is it a combination of catchments, and if yes, which?

Depending on the assessment, our study area varies; the Figure contains our region of main interest (focus region) and corresponds to the light red box in Figure 3 (LReg). We will avoid the term "study area" and make it clearer in the text, which area is meant depending on the analyses.

**Ll. 93-100:** I am not sure how helpful this paragraph is for the audience. If specific districts or municipalities are important in a spatial context, they should be included in a map. If the main map is too crowded or its scale to small, you can provide another inset or sub-plot which e.g. focuses on the administrative structure within e.g. the Ahr and the Erft catchments.
The names are often used only once in the text (e.g. picture reference or "special report"). We will add the city of Bad Neuenahr-Ahrweiler to the figure and shorten most of the text here. Small additions to the names should be sufficient for orientation, since all are located along the rivers Ahr / Erft.

**Ll. 125-130:** Why is a weather forecast model used to analyse geopotential patterns and precipitable water? Why not use an analysis? The section header says "Weather forecast and analysis data", but the section does not describe any analysis data.
We will renew the Figure 2a and will use the ICON model. Please note that in the new version we will shift (and extend) the information on ERA5 in this subsection.

**Ll. 146:** Again, you mention the study area with regard to the selection of river gauges. Please explicitly specify, e.g. in Fig. 1, what you consider as the study area.
Please see our answer above; we will specify this in more detail in the new version.

You also specify a lot of selection criteria; but why did you not just use, in the context of your study, all river gauge data that were available to you?
For this study, we were in the fortunate position to collaborate with representatives of several water administrations and water agencies. The number of gauges and gauge data thus available to us was large, and it would have been neither possible nor helpful for the reader to include and discuss all these data in the manuscript. We therefore decided to restrict ourselves to a small yet representative (in terms of spatial coverage, catchment size etc. along the criteria mentioned in the manuscript) set of gauges. We do believe that the selected set of 10 gauges is a good compromise in this respect. We will add one sentence for clarification.

**Ll. 153 ff.:** I think the reconstruction of water level or discharge is an important methodological feature and should hence be described or adequately referenced.
We fully agree with the referee that the reconstruction of water levels (e.g. from debris lines) and discharge (from water levels) is an important aspect of the reconstruction of the event, especially as the involved uncertainties are large. A key element in this respect is that there is no single institution doing this, based on a single method. Rather the individual water authorities operating the gauges, or responsible for particular rivers, do this, using a variety of approaches based on available data, expertise, finances, and interest. It is therefore next to impossible to provide a detailed yet complete description of how the gauge data were reconstructed. Nevertheless, in lines 153ff we mention that the reconstructions were done by the data providing water authorities, and name them,

and provide in Sect.3.2 more details about the manner of reconstruction and related uncertainties. E.g. Lines 375 ff, 406 ff, 432 ff. Nevertheless, in the revised version we will add a short section in the data section, bundle the mentioned information and bring it forward to this section and discuss the general problem regarding this.

**Ll. 180 ff.:** You are using ERA5, but you did not specify ERA5 as data used in this study in section 2.1. Currently, ERA5 is mentioned in section 2.2.1 - admittedly quite briefly; we will move (and expand) the information in Section 2.1.2.

**Ll. 195:** you say *"for example, [...] daily precipitation totals"*. Which variables, apart from that, were subject to extreme value statistics?
We will rephrase this as we are only looking at precipitation data in the study ("such as in the following...").

**Ll. 239:** *"very high values of total precipitable water of more than 40 kg m−2 were reached, which occur only very rarely."* - how rarely?
We will perform a quantitative assessment using available historical radiosonde data for northern Germany to provide a better context.

**Ll. 256 ff.:** The authors end the section with the sentence *"Heavy precipitation associated with quasi-stationary low pressure systems, their fronts, or convective systems located on the western flank of persistent blocking systems is common in Europe during summertime, so the large-scale situation is not unusual"*. So what do we actually learn from the section *"Synoptic overview and atmospheric characteristics"*? How does it help us to understand how the event unfolded?
In terms of (extreme) weather events, it is always important to consider the large-scale context or processes and mechanisms, and to mention factors that contributed significantly (e.g., air mass transport, large-scale lifting processes, blocking). We will rewrite the passage to better emphasize the importance of the large-scale processes in relation to the rainfall (important precondition, but not unique).

**Ll. 262:** How do you know the presence of embedded convection?
Based on radar imagery and recorded precipitation intensities, it can be assumed that the precipitation was convection amplified. We will also cite another study here (Kreienkamp et al., 2021).

**L. 269:** please quantify what you mean by *"major part"*.
We will add this information in the revised version.

**Fig. 3:** Why not zoom all maps into the study area? I do not find it helpful to show all of Germany. For the API, it would also help to show return periods in order to appreciate whether/how the API was anywhere near unusual for this event.
We will focus on the LReg region in the revised version in the figure. In addition, we will examine how soil moisture (or API) behaves in a historical context.

**Fig. 4:** for the sake of comparability, please use the same x-axis for subplots a and b;
A unification is not suitable, since we consider different time scales (or the information density is different): EFI starts already on 9.7.; additionally, we have in (a) from 13.7 even 3-hourly intervals; EFI, however, is only available 12-hourly.
I assume the black dashed line (observed reference from RADOLAN) is the spatial average - but over which region? Please specify in the figure caption.
Also LReg; It's already mentioned in the figure caption: *"24 h precipitation totals over LReg"*.

**Ll. 319 ff.:** At some point, you need to provide a bit of context on the interpretation of the EFI - also quantitatively: How frequent is an EFI exceedance > 0.8 in the region, hence how well does it signify the potential for such a singular event?
This is not easy to answer, because we can't do statistics (no data). An EFI of 0.8 means that 80 % of the ENS-members exceed the maximum value of the Mean-climate, which certainly does not occur too often. The EFI was specially developed so that "extremes" can be predicted and not that with a small shower a big warning starts.

**Ll. 343:** *"twice the climatological mean"* is not very helpful, in my opinion, as the reader is not informed about the statistical distribution of API values in the area. It would be more informative to provide a (extreme value) statistical assessment in terms of frequency.
We try to specify this better (maybe as suggested by return periods).

**Ll. 345 ff.:** *"In the southern parts of the Eifel, the Ardennes in the north-west, and in the north-east of the study area in the Wupper region, generally less than 10 mm of soil water storage were still available for infiltration. In the remaining regions, free soil water storage was larger, but still below average, ranging mainly between 10 and 30, sometimes 75 mm."* On which basis are these statements made? The API does not allow for such an assessment (by the way, since the API is only computed from precipitation, it is basically a meteorological quantity, not a hydrological one).

Thank you for pointing this out. The API indeed does not allow statements on available soil water storage. The statements were made based on Fig. 2 in Junghänel et al. (2021) (see below). We will make this clear in a revised version of the manuscript.

[Figure]

**Abb. 2**: *Freier Bodenwasserspeicher auf einer 60 cm dicken Bodenschicht unter Gras und einem realistischen Boden nach BÜK 1000 vor der Unwettersituation am 12.07.2021. Quelle: DWD, Agrarmeteorologie*

More importantly, in the context of your study and its stated objective, it would be crucial to assess the extent to which drier soils (e.g. average soil moisture in July) could have been able to retain substantial amounts of water and hence significantly reduce the hydrological response.

We agree with the referee that it would be interesting to investigate in 'what-if' scenarios how the event would have unfolded under different antecedent conditions. However, the main focus of this paper is a multidisciplinary documentation and reconstruction of the event as it was rather than scenarios. In the companion paper, we do such studies in the form of scenarios related to global warming (Pseudo Global Warming studies). We therefore prefer to not include such a study into the current manuscript.

**Ll. 368 ff.:** Apart from the fact that peak flow was exceptionally high with a steep rise, I am not really sure what to learn from the entire section about "Ahr, Kyll, and Prüm river basins". I think it would be a perfect opportunity to analyse why quite a similar rainfall in the headwaters lead to different event amplitudes in the different basins. These lines (ll. 368 ff.) aim at that, but I have the impression that the statements are rather based on speculation.

Our aims of reporting the course of the flood event at the rivers Ahr, Kyll and Prüm are twofold: First, to show that the 2021 event was not a local one, affecting only a single river, but rather it affected a large region and many watersheds. Second, that for all rivers, the course of the event was quite similar and characterized by steep rise and exceptional peak water levels. Owing to local catchment size, river and floodplain geometry at the gauges, the peak water levels and flows naturally differ, but overall they are quite similar in the sense that they are well beyond the 100-year flood. From these rivers, the peak water levels at the Ahr river stand out, as indicated by the very high peak factors, and we also give an explanation for this in lines 368 ff (steeper topography and narrower valleys than in at Kyll and Prüm). This is further detailed in Sect 3.3. So we would like to maintain that our statements are not speculative, but based on comprehensible arguments.

**Fig. 5 and ll. 373 ff.:** How were the time series reconstructed (dashed lines)? I understand that peak water level and hence peak flow can be reconstructed from debris lines etc., but how about the recession? How can we explain the high water level at gauge Schönau after the peak, while discharge had already recessed to normal levels?

On the general reconstruction of the waterlevel and discharge time series, please see our reply to RC LI. 153 ff.:.

In particular about the recession limb: Sometimes, water level recordings were not available during the flood, but for a few points in time, water level reconstruction was possible from photos available after the flood, plus temporal interpolation. In other cases, recession limbs could be estimated from recession limbs from upstream gauges with observations. Waterlevel recordings or estimates could then be used to estimate discharge. For this, often pre-event W-Q relations could not be used because of substantial changes of the river cross-section (erosion or deposition). This is for example the case for gauge Schönau, where deposition lead to continuously high water levels after the flood despite declining discharge. Such effects were detected by the responsible water authorities and considered in the reconstruction of discharge.

We will briefly address this in the new data section (see above).

**Ll. 456 ff. / section 3.3:** many statements in section 3.3 are not identifiably based on scientific data - either observation or models -, but seem to be speculative, hear-say, or partly reproducing textbook knowledge. The actual scientific contribution of this section remains unclear to me, especially in comparison to the paper of Dietze et al. (2022) which is widely cited by the present study.

Somehow, the reviewer is right. However, we needed to frame our analysis in textbook knowledge to be able to further show point out the gap between these and normal flood management procedures. To obtain meaningful quantitative data is almost impossible (not casuistic but enough data and through a minimum sampling scale), and to go for a modelling approach would take an effort not compatible with this manuscript. With the analysis of the sparse data which is available in terms of morphology changes in the valley, we aimed at (i) clearly demonstrate the importance of the hydro-morphodynamic processes in the enhancement of the valley hazard; (ii) to clearly show how the anthropic modification of the catchment enhances the flood hazard; (iii) to identify and explain how morphology (natural or anthropic) singularities of the valley become focus of hazard during floods; (iv) how hydrodynamic and geomorphic processes explain the damages which we observed in a post event analysis. We understand that the objectives of the paper by our colleagues Dietze et al. (2022) has somehow commonalities, however we (i) go further in aspects related to the understanding of the anthropic modification of the valley (mainly of urban and industrial character), (ii) we present complementary and novel information and cases and (iii) our interpretation fills some gaps or questions the interpretations by our colleagues.

In the next version of the paper, we will avoid or clearly assume speculative statements, and we will identify better our common points and highlight differences with previous analysis of the event by colleagues.

**Fig. 9:** I think that the scale of this figure is not really helpful for appreciating the inundated area.
Please keep in mind that this is a general map and that the Ahr is only 2 to 4 m wide. Nevertheless, we will try to optimize the map a bit more (trim and rotate).

**Ll. 589 / section 4.1:** I think the issue of rapid inundation mapping is an important research topic, but I do not see how it is relevant in the post-event analysis almost one year after the fact.
see answers above (new section and research questions)

**Ll. 630 ff. / section 4.2:** The same applies to section 4.2: Why is the rapid damage estimation process important? In the context of this study, shouldn't the best possible estimates of damages and damage processes be used? Furthermore, section 3.3 emphasised very much the role of debris/sediment/hydro-morphodynamics for damage processes, but I do not find this issue in section 4.2. Isn't this damage process chain one of the key properties of the July 2021 event?
see answers above (new section and research questions)

**Ll. 700 ff. / section 4.3:** This section is mainly based on the number of traffic reports and the number of affected railroad sections, irrespective of the severity of disruption. Doesn't that limit the meaningfulness and the comparability to the 2013 event, given that it should matter whether a road/bridge is just disrupted or whether it actually disappeared? Maybe the persistence of the disruption for specific sections/lines could be interpreted as a proxy for severity, but there should be a reference from other events to compare to.
It would certainly be interesting to examine this in more detail; but unfortunately we do not have any information on the severity, so a comparison is difficult; we can only make statements about this event and are limited in the details. We will emphasize this in more detail when we compare it with 2013.

**Fig. 11b:** In total, I find subplot b not very informative, particularly since most of the specific road and railway numbers are not referenced in the main text. Wouldn't a simple plot of affected sections/lines over time be more informative and concise?

In accordance with your comments at the beginning, we will shorten this section and move the figure to the supplement, as the main statements can be concluded without the figure.

**Ll. 744 ff:** *"The July 2021 flood in western Germany and neighbouring regions was one of the five most severe and expensive natural catastrophes in Europe in the last half century"* - what is the source of that statement?
This is our result; see section 4.2. line 696. We will emphasize this again better in Section 4.2.

How important is the consideration of EFI in the context of your study, given the apparently high skill of the ICON models?
The point here is that not only the ICON but also other models (here: ECMWF-ENS) predict the extreme event. This again speaks for the general good predictability of the event (from the meteorological side) and that not only one model had this event on the screen. We will specify this.

**L. 763:** *"However, because of the high political relevance, the failure of the warning chain was not discussed here."* I understand, but why bring it up then? Could in this case the topic of predictability also be dropped from the manuscript?
Based on our expertise, we have the opportunity to show that the meteorological models predicted the extreme event quite well. We cannot go into more detail on the further warning chains, as this is beyond our capabilities. However, it is important to mention this aspect in the outlook of the paper.

We will rephrase the sentence: "Despite the early prediction of the extreme event, the warning chains did not work properly. However, the analysis of the failures is not the subject of this article."

**L. 809:** *"Remote sensing of rapidly available imagery from social media, television, and news media [...]"* - I don't understand what that means.
We will rewrite the sentence.

**Ll. 817 ff.:** *"[...] our still insufficient knowledge must be further improved through more dedicated research"* - is it really necessary to spell out the obvious?
We will rewrite the sentence.

**Ll. 823 ff.:** Is this item a new conclusion drawn from this study? Wasn't this already pointed out by Roggenkamp and Herget (2022)?
Based on the work we have done, we both came to this conclusion (cf. Schäfer et al., 2021); in the end, this point cannot be emphasized often enough. We will refer to Roggenkamp again at this point.

**Ll. 849 ff.:** Sure, but how??
We will include in the revised version of the paper our suggestions, which include remote sensing and video analysis techniques, which if well calibrated can be used as non-intrusive (less accurate) gauging stations in the case of similar extreme events.

**Ll. 853 ff:** How is this a conclusion from the present study?
see answers above (new section and research questions); we will rewrite this.

**Ll. 876 ff.:** Please see my above comment: I think it would be really helpful to publish the data along with this manuscript.
see answer above.

**Technical comments**

Thank you for your carefully reading; unless otherwise indicated, we will consider all the comments below.

**L. 160:** Please replace "flood plain" by "inundated areas"
Thanks, we will change this to "inundation areas" in consistence with the whole text.

**L. 189:** were used

**L. 191:** 95th percentile

**L. 210:** suitability instead usability

**L. 224:** second "decade"?
We will rewrite this.

**Ll. 272 ff.:** *"The long-term average for the month of July at this station is 69 mm (1981-2010); thus, in just a few hours, the rainfall added up to more than twice the usual monthly precipitation."* - This kind of statement is unnecessary.
We will delete the second part.

**L. 307:** as early as

**L. 355:** replace same by similar

**L. 580:** replace "the hardest affected location" by "the location affected most severely"

**Fig. 10:** It impossible to distinguish EMSR517_v1 from EMSR517_v3
We will improve the readability of the figure.

**Fig. 11:** the resolution is very low.
Due to the comments above, we have decided to no longer use the figure in the main document (see above).

**L. 762:** delete "the occurrence"

**L. 831:** an extraordinary

**L. 837:** the term "re-forecast" sounds weird - isn't "hindcast" the correct term?
Both is common; for example, re-forecast is commonly used by the ECMWF:
https://www.ecmwf.int/en/forecasts/documentation-and-support/extended-range/re-forecast-medium-and-extended-forecast-range
https://www.ecmwf.int/en/forecasts/datasets/set-vi

---

## Author Comment (AC2)

**Reviewer #2**

**General comments**

We thank the reviewer for his/her time and comments, corrections, and suggestions.

The paper "A multi-disciplinary analysis of the exceptional flood event of July 2021 in central Europe. Part 1: Event description and analysis" by Mohr et al. gives an overview of the flood event last year, with a special focus on hydrological, and hydro-morphological processes and mechanisms. The paper describes very well the event across various physical disciplines. The complex interaction between those could be analyzed in more detail. This was also a comment of the other reviewer. We will better address this aspect in the revised version and link better the results between the individual disciplines. In addition, we will introduce a new section that will be a "synopsis" and thus better bring together the results of the different disciplines. Please see also our answers to Reviewer 1 for more details.

The aspects of social science regarding the flood event are not addressed, at least the paper should underline of refer to the high importance of risk culture (e.g. risk awareness, risk communication).
This is more difficult to take into account; first, the other reviewer points out that our work should be more stringent and we should concentrate more on our own results with clearly defined questions (and not be too broad in the text). Second, this aspect (risk culture) is not our objective of the study, as we did not do own research on this topic. However, we can address this aspect a bit more in the outlook, as there are also publications by others on this topic to which we can then refer.

**Specific comments**

L 1; L 16; L 69; L334, L744: Please think about if you want to use the term "natural disaster". There is no disaster without human interference, so it's never something "natural" Have a look at #nonaturaldisaster: https://www.nonaturaldisasters.com/
That is of course correct; we will adapt that (either "natural hazard" or only "disaster").

L 23: Figure 1 is mentioned here for the first time, but Figure 1 is currently in L 116. Why so far away?
We can move the figure to the introduction, but in the end the publisher's typesetters will decide the position.

L 35: displaced people? Yes, we will add "people".

L 39: in the meantime, flood hazard maps are updated see Roggenkamp & Herget. I would rather write the existing maps before and during the flood
Yes, that's right, we know that these have been updated in the meantime; therefore we will use the suggested formulation.

L 107: see below – could you describe below more specific please?
In fact this reference is irrelevant (or unnecessary), we will delete it.

**Technical corrections**

Thank you for the carefully reading; unless otherwise indicated (e.g., numbers), we will consider these technical corrections.

L105: one hour, but can reach up to one minute → Numbers from one to twelve are written out
According to the NHESS guidelines (https://www.natural-hazards-and-earth-system-sciences.net/submission.html), this would in general be correct (..."use words for cardinal numbers less than 10"…); however, for items that are "units of time or measure", this is not the case, and here this is a time unit.

L 38: only one week
See above

L63: six month
See above

L73; L74: (e.g. ….)

L74: erosion, and

L128: used by

L129: In its global uniform resolution configuration it is run twice daily → check the grammar
We will include a comma

L307: as early as

L310: were predicted more than two days
See above

L320: two days
See above

L339: - namely soil wetness → missing spaces

L344: three weeks
See above

L371: erosion, and

L372; L389f; L401, L484: - …. → missing spaces

L408f: the peak flow

L523: floods are?

L632: infrastructural, and

L762: two days
See above

L857: This helps to mitigate associated adverse effects -> missing . in the end

---

## Author Response (AR1)

**Reviewer #1**

**General comments**

In their manuscript *"A multi-disciplinary analysis of the exceptional flood event of July 2021 in central Europe. Part 1: Event description and analysis"*, Susanna Mohr and colleagues provide an overview of the disastrous July 2021 flood event.

We thank the reviewer for his/her time, the very careful and thorough review, and the many suggestions. We are aware that he/she took a lot of time.

The authors have done an impressive job in collecting and compiling information on various aspects of the July 2021 flood, and the sheer effort behind this needs to be appreciated.

Having said that, the objectives and specific research questions related to this study remain unclear to me. In ll. 74 ff. of the introduction, the authors state that *"[...] the objective of this two-part study is a multi-disciplinary assessment of the entire process chain of the July 2021 flood in central Europe - from causes to impacts to historical classification and climatological context [...] While Part 1 focuses on the description of the event across various disciplines (meteorological, hydrological, hydro-morphological, economic) [...]"*. This is not a research question, and nowhere in the paper, the authors specified their idea of a *"multi-disciplinary assessment"*. In ll. 754-755 of the conclusions, they state that *"this paper examined the complex interactions among meteorological, hydrological, hydraulic, and geomorphological processes and mechanisms that led to the extraordinary flood [...]"*. I find this statement difficult to confirm: instead, the manuscript largely remains a *description* from different sub-disciplines (meteorology, hydrology, hydro-geomorphology, impacts/damages), listed one after another, but mostly not related to each other in terms of an *analysis*. This is a challenge that many papers have to cope with when they aim to provide a holistic assessment of an extreme event - I noticed a similar referee comment on the paper of Caldas-Alvarez et al. (2022) about the Berlin 2017 event in this same special issue (link).

While it is, in the direct aftermath of such a disaster, valid and required to focus on the rapid compilation of information and data, and to make such compilations available to decision makers, the research community, and the general public, I wonder whether we should have, by now, reached a phase in which the research community should find more well-defined modes of event analysis. Then again, I am aware that putting together such a manuscript requires a lot of time, and that the processes for this probably had already started in 2021. My overall recommendation is that the authors take a step back and re-evaluate the purpose of this paper. What is your scientific objective, aside from compiling as much as we know about the event? I think it would help the paper very much to, after a brief synopsis of the event, identify maybe two or three important and *specific* research questions (e.g. with regard to specific interactions), and tell the story along these questions - instead of just listing one discipline after the other. Many aspects of the paper are quite interesting in itself, but maybe not required for a holistic view? Do they have to be a part of this study?

We revised the content and structure of the paper. Most notably, this includes defining two new research questions that better reflect our motivation of the study and our results, respectively. The two new research questions (RQs) are:

1) What were the hydro-meteorological causes of the July 2021 flood and what interactions and impacts were observed? What made the flood so exceptional?
2) What additional information can be generated directly in the aftermath of an extreme flood event to support disasters management and how reliable are first estimates? (context early response)

Another important change is the inclusion of a new section that is a "synopsis" to better bring together the results of the different disciplines (to address the criticism that the results of the different disciplines are just "attached" to each other). In addition, this section is also devoted to RQ2. In addition, we included a new figure, a timeline of what happened and what was done in the aftermath (concerning our study). We think that with the new section we better highlighted the "multidisciplinary assessment" and the importance or possibilities of near-real-time forensic disaster analysis.

Regarding the "test reduction" aspect: based on our new RQs, we believe that we addressed all the previous findings (from the first version) and that they should still be part of the study; however, we are aware that we had to cut within sections and did to reduce significantly (e.g. reducing introduction, old Fig. 12 is now part of the supplement, deleted equations).

For example, I found it difficult to understand why we need the section on the "synoptic overview and atmospheric characteristics". And the predictability of heavy rainfall (weather forecast analysis) would be relevant only if the authors had actually investigated the failure of the early warning chain and the resulting implications on the impacts in terms of damage and loss of lives. Also the interaction between discharge dynamics, hydro-morphodynamics and impacts is not yet sufficiently elaborated.

In terms of (extreme) weather events, it is always important to consider the large-scale context or processes and mechanisms, and to mention factors that contributed significantly (e.g., air mass transport, large-scale lifting processes, blocking). For example, atmospheric blocking was crucial that the Baltic Sea could be warmed so significantly and served as a source for the air masses. We rewrite the subsection (and also reduced some passages). The last two points are now better addressed in the new section 5 (synopsis).

I understand that it is challenging to revise the paper along these lines, but I am confident that the authors will find an adequate way. Is it helpful to recommend that the paper should be much shorter? It almost took me three days just to work my way through it. I think the length of the paper could be easily reduced by at least a third and I hope this will help separating what is relevant from what is not.

Although we did not delete content from the paper, as all make important contributions in the context of the two new research questions, we deleted individual passages of text or sentences in many places.

One more comment on the issue of multi-part papers: Surely it is up to the editorial team to assess whether a multi-part publication is warranted. Personally, I have never really understood the need for multi-part papers. A paper should be self-sustained, and *evaluated* as an individual piece of scientific work. This is also the basis for this review. Of course, papers can and should refer to each other and build upon each other, but in my view, that does not require an explicit multi-part approach. In the present context, the multi-part approach supports the impression that PART1 is more about compiling "everything we know" instead of asking and addressing well-defined research questions.

We think that through the revision of the present paper (including new research question) and the meanwhile published PART2, the interplay between the two publications has become more clear:
Ludwig, P., Ehmele, F., Franca, M. J., Mohr, S., Caldas-Alvarez, A., Daniell, J. E., Ehret, U., Feldmann, H., Hundhausen, M., Knippertz, P., Küpfer, K., Kunz, M., Mühr, B., Pinto, J. G., Quinting, J., Schäfer, A. M., Seidel, F., and Wisotzky, C.: A multi-disciplinary analysis of the exceptional flood event of July 2021 in central Europe. Part 2: Historical context and relation to climate change, Nat. Hazards Earth Syst. Sci. Discuss., https://doi.org/10.5194/nhess-2022-225, in review, 2022.

**Specific comments**

**Is this a research article or a review?**

To be honest, I am also not sure which type of manuscript I am dealing with. Over large parts, the manuscript more resembles a review paper instead of a research article. It struck me that there is no formal "results" section (instead "event description and analysis").

This is not a fundamental issue in itself; however, it again points to the fact that there are no specific research questions and hence no specific results to address these.

We suspect that the introduction in the first version initially gives the impression of a review paper, as our motivation was originally to refer to previous activities/studies on the flood event in general. In the revised version, we deleted this paragraph completely and adapted the introduction more appropriately to our new defined research questions (RQs). We also believe that the reformulation of the RQs also makes clearer that our study is a "research article." Section 3 and Section 4 correspond to our results sections, and the new Section 5 brings together the results and addresses our new research questions, respectively.

Furthermore, the content in section 3 ("event description and analysis") is largely not based on the data and methods which have been described in section 2 ("data and methods"). I would like to discuss this in detail: Section 2.1 (data) documents precipitation data, atmospheric model data, river gauge observations (not the data/methods on reconstruction of water levels/discharge), Sentinel1/2 data (for inundation mapping), and traffic data (reports on road and railway disruptions). Section 2.2 (methods) documents the trajectory analysis (for moisture source analysis), extreme value statistics for precipitation, and the computation of the antecedent moisture index. Together, section 2 incompletely addresses the methods and data that were used to put together section 3; in section 3.2, for instance, reconstructed water levels/discharges play an important large role; section 3.3 (hydro-morphodynamic processes) is almost entirely unrelated to data and methods documented in section 2; and section 4 (impacts and consequences), too, is based on methods and data sets (e.g. aerial/media footage, loss models, insurance data)

which were not mentioned in section 2 (except the Sentinel data and the reports on traffic disruptions).
We revised Section 2 and added missing information (or moved it to the front from the back of the paper). The description of the ERA5 data can now be found in Section 2.1; also the methods for reconstruction of water levels/discharges (Section 2.3). The procedure to determine the inundation areas (partly described in section 4.1. in the first version) can now be found in a separate sub-section (2.4) and also a sub-section (2.7) regarding the loss modeling was added.

Altogether, I understand that the variety and mass of methods and data that section 3 is based upon is almost impossible to describe in section 2. The reason for that is that much of the content shown in section 3 is not really based on the application of data and methods in the context of study, but rather a compilation and synthesis from other recent studies about the July 2021 event, namely Fekete and Sandholz (2021), Dietze et al. (2022), Thieken et al. (2022), Schäfer et al. (2021), Apel et al. (2022), BM (2022), Roggenkamp and Hergert (2022). Hence, I would like to ask the authors to clarify, from the beginning, how they combine the original analysis of data and methods with the results of other studies in the context of the paper.

We think that due to the reorganization of the text (incl. section 2) and the introduction of the new section (synopsis), we succeeded better in separating our own work from that of others. Please note that Schäfer et al. (2021) is our own work, which is the basis for this publication (however, it is only a technical report, which has not been peer-reviewed).

Furthermore, I suggest that the authors publish the data which they used for this study in a single dedicated and documented data set to accompany this manuscript as an asset, even if parts of this data are available elsewhere (like the precipitation data or the Sentinel data). That way, they would not only make the original contribution of this study more transparent, but it would also be a valuable service to the research community. I understand that this will require some coordination with the source institutions, e.g. for the river gauge observations or the railway disruption data. Still, I think it'll be worth the effort.
Unfortunately, it is not possible to publish all the data/time series used, as we do not have permission for all of them. However, in the section "Code and data availability" we indicate from where the data can be obtained.

But we published the datasets that we produced or generated ourselves. The inundation data and the air mass trajectories can now be found here: https://zenodo.org/record/7357466#.Y4DbYX2ZM-V (doi is in requested).

**Text is reproducing figures and tables**

Very often, the content of figures and tables is reproduced/reiterated in the main text. I think the authors should trust more in the information content of their figures, and if they don't, figures should be made more concise.

With respect to Sect. 3.2 (Hydrological aspects), we would like to maintain that the joint discussion of the flood evolution over time in the text, and comparing them to historical and statistical peak flows, is a useful addition to the hydrographs as shown in Fig. 5 and listed in Table 1. Neither does the figure alone convey the points discussed in the text, nor can the text stand alone, without the figure. We therefore prefer keeping the text, in combination with the figure as is.

**Introduction and conclusions sections do not sufficiently frame the study**

The introduction reflects the general issue of this paper: instead of using the introduction to systematically develop and justify specific research questions, it appears to string together previous studies in a rather unrelated fashion.
We revised (and shortened) the introduction, including our research questions. Now the text focuses less on a collection of existing work on the flood in 2021; in contrast, the text addresses the severity, aims of forensic disaster analysis, importance & problems regarding early response, more fitting to our new research questions.

Similarly, the section "discussion and conclusions" (four pages!) provides a long list of statements, and for many of them, it is not really clear how they are based on the results presented in this study and how they relate to a study objective.

We revised the conclusion.

**Other comments**

**Fig. 1:** Please make sure that the border of Luxembourg is visible behind the river Sauer; I am not sure why the catchments are hatched - it does not improve the readability of the map. Furthermore, please show the catchment boundaries for all river catchments discussed in this paper (see section 3.2).
We revised the readability of the map and added the city of Bad Neuenahr-Ahrweiler (to be able to delete the

paragraph at the beginning of Section 2). To show all catchment areas would make the map confusing and go beyond the context of the paper; therefore, we limit the revised version to our primary focus area and show only the catchment of the Ahr.

**L. 90:** While we see an area in the map, it remains unclear what signifies the "study area". Which parts of the area are actually *studied*? Is it the area in which specific precipitation totals were exceeded? Is it a combination of catchments, and if yes, which?
Depending on the assessment, our study area varies; the Figure contains our region of main interest (focus region) and corresponds to the light red box in Figure 3 (LReg). We avoided the term "study area" and make it clearer in the text, which area is meant depending on the analyses.

**Ll. 93-100:** I am not sure how helpful this paragraph is for the audience. If specific districts or municipalities are important in a spatial context, they should be included in a map. If the main map is too crowded or its scale to small, you can provide another inset or sub-plot which e.g. focuses on the administrative structure within e.g. the Ahr and the Erft catchments.
We deleted the whole paragraph and added the city of Bad Neuenahr-Ahrweiler to Fig 1 for orientation. Furthermore, Figure 9 (now 10) was revised and includes now a part of the location names (those along the Ahr), thus this helps now also for orientation.

**Ll. 125-130:** Why is a weather forecast model used to analyse geopotential patterns and precipitable water? Why not use an analysis? The section header says "Weather forecast and analysis data", but the section does not describe any analysis data.
We renewed the Figure 2a using ERA5 data for both subfigures.

**Ll. 146:** Again, you mention the study area with regard to the selection of river gauges. Please explicitly specify, e.g. in Fig. 1, what you consider as the study area.
Please see our answer above; we rewrote this in the new version.

You also specify a lot of selection criteria; but why did you not just use, in the context of your study, all river gauge data that were available to you?
For this study, we were in the fortunate position to collaborate with representatives of several water administrations and water agencies. The number of gauges and gauge data thus available to us was large, and it would have been neither possible nor helpful for the reader to include and discuss all these data in the manuscript. We therefore decided to restrict ourselves to a small yet representative (in terms of spatial coverage, catchment size etc. along the criteria mentioned in the manuscript) set of gauges. We do believe that the selected set of 10 gauges is a good compromise in this respect. We added one sentence for clarification.

**Ll. 153 ff.:** I think the reconstruction of water level or discharge is an important methodological feature and should hence be described or adequately referenced.
We fully agree with the referee that the reconstruction of water levels (e.g. from debris lines) and discharge (from water levels) is an important aspect of the reconstruction of the event, especially as the involved uncertainties are large. A key element in this respect is that there is no single institution doing this, based on a single method. Rather the individual water authorities operating the gauges, or responsible for particular rivers, do this, using a variety of approaches based on available data, expertise, finances, and interest. It is therefore next to impossible to provide a detailed yet complete description of how the gauge data were reconstructed. Nevertheless, in lines 153ff we have already mentioned that the reconstructions were done by the data providing water authorities, and name them, and provide in Sect.3.2 more details about the manner of reconstruction and related uncertainties. E.g. Lines 375 ff, 406 ff, 432 ff.
Nevertheless, we added now a short paragraph in the data section, bundled the mentioned information, and discussed the general problem regarding this.

**Ll. 180 ff.:** You are using ERA5, but you did not specify ERA5 as data used in this study in section 2.1.
We now added the information about ERA5 in the Section 2 at the beginning.

**Ll. 195:** you say *"for example, [...] daily precipitation totals"*. Which variables, apart from that, were subject to extreme value statistics?
This subsection was shortened to reduce the overall text as suggested, since the method is quite well-known and we have already cited appropriate publications that describe the procedure (including the equations) very well.

**Ll. 239:** *"very high values of total precipitable water of more than 40 kg m−2 were reached, which occur only very rarely."* - how rarely?
We deleted this comment.

**Ll. 256 ff.:** The authors end the section with the sentence *"Heavy precipitation associated with quasi-stationary low pressure systems, their fronts, or convective systems located on the western flank of persistent blocking systems is common in Europe during summertime, so the large-scale situation is not unusual"*. So what do we actually learn from the section *"Synoptic overview and atmospheric characteristics"*? How does it help us to understand how the event unfolded?

In terms of (extreme) weather events, it is always important to consider the large-scale context or processes and mechanisms, and to mention factors that contributed significantly (e.g., air mass transport, large-scale lifting processes, blocking). We rewrote the passage to better emphasize the importance of the large-scale processes in relation to the rainfall (important precondition, but not unique).

**Ll. 262:** How do you know the presence of embedded convection?

Based on radar imagery and recorded precipitation intensities, it can be assumed that the precipitation was convection amplified. We add a comment.

**L. 269:** please quantify what you mean by *"major part"*.

We rewrote this.

**Fig. 3:** Why not zoom all maps into the study area? I do not find it helpful to show all of Germany.

Figure 3 was revised and shows now a cutout of the relevant region.

For the API, it would also help to show return periods in order to appreciate whether/how the API was anywhere near unusual for this event.

We added a new Figure (Fig. 5), which shows the frequency distribution of spatially averaged daily API values over LReg based on HYRAS (1951\,--\,2015). Furthermore, point-based analyses of the return periods of the API were included in the supplement.

**Fig. 4:** for the sake of comparability, please use the same x-axis for subplots a and b;

A unification is not suitable, since we consider different time scales (or the information density is different): EFI starts already on 9.7.; additionally, we have in (a) from 13.7 even 3-hourly intervals; EFI, however, is only available 12-hourly.

I assume the black dashed line (observed reference from RADOLAN) is the spatial average - but over which region? Please specify in the figure caption.

Also LReg; It's already mentioned in the figure caption: "24 h precipitation totals over LReg".

**Ll. 319 ff.:** At some point, you need to provide a bit of context on the interpretation of the EFI - also quantitatively: How frequent is an EFI exceedance > 0.8 in the region, hence how well does it signify the potential for such a singular event?

This is not easy to answer, because we can't do statistics (no data). An EFI of 0.8 means that 80 % of the ENS-members exceed the maximum value of the Mean-climate, which certainly does not occur too often. The EFI was specially developed so that "extremes" can be predicted and not that with a small shower a big warning starts.

**Ll. 343:** *"twice the climatological mean"* is not very helpful, in my opinion, as the reader is not informed about the statistical distribution of API values in the area. It would be more informative to provide a (extreme value) statistical assessment in terms of frequency.

We added a new Figure (Fig. 5), which shows the frequency distribution of spatially averaged daily API values over LReg based on HYRAS (1951\,--\,2015). Furthermore, point-based analyses of the return periods of the API were included in the supplement.

**Ll. 345 ff.:** *"In the southern parts of the Eifel, the Ardennes in the north-west, and in the north-east of the study area in the Wupper region, generally less than 10 mm of soil water storage were still available for infiltration. In the remaining regions, free soil water storage was larger, but still below average, ranging mainly between 10 and 30, sometimes 75 mm."* On which basis are these statements made? The API does not allow for such an assessment (by the way, since the API is only computed from precipitation, it is basically a meteorological quantity, not a hydrological one).

Thank you for pointing this out. The API indeed does not allow statements on available soil water storage. The statements were made based on Fig. 2 in Junghänel et al. (2021) (see below). We rewrote this part in the new version.

[Figure]

Abb. 2: Freier Bodenwasserspeicher auf einer 60 cm dicken Bodenschicht unter Gras und einem realistischen Boden nach BÜK 1000 vor der Unwettersituation am 12.07.2021. Quelle: DWD, Agrarmeteorologie

More importantly, in the context of your study and its stated objective, it would be crucial to assess the extent to which drier soils (e.g. average soil moisture in July) could have been able to retain substantial amounts of water and hence significantly reduce the hydrological response.

We agree with the referee that it would be interesting to investigate in 'what-if' scenarios how the event would have unfolded under different antecedent conditions. However, the main focus of this paper is a multidisciplinary documentation and reconstruction of the event as it was rather than scenarios. In the companion paper, we do such studies in the form of scenarios related to global warming (Pseudo Global Warming studies). We therefore prefer to not include such a study into the current manuscript.

**Ll. 368 ff.:** Apart from the fact that peak flow was exceptionally high with a steep rise, I am not really sure what to learn from the entire section about "Ahr, Kyll, and Prüm river basins". I think it would be a perfect opportunity to analyse why quite a similar rainfall in the headwaters lead to different event amplitudes in the different basins. These lines (ll. 368 ff.) aim at that, but I have the impression that the statements are rather based on speculation.

Our aims of reporting the course of the flood event at the rivers Ahr, Kyll and Prüm are twofold: First, to show that the 2021 event was not a local one, affecting only a single river, but rather it affected a large region and many watersheds. Second, that for all rivers, the course of the event was quite similar and characterized by steep rise and exceptional peak water levels. Owing to local catchment size, river and floodplain geometry at the gauges, the peak water levels and flows naturally differ, but overall they are quite similar in the sense that they are well beyond the 100-year flood. From these rivers, the peak water levels at the Ahr river stand out, as indicated by the very high peak factors, and we also give an explanation for this in lines 368 ff (steeper topography and narrower valleys than in at Kyll and Prüm). This is further detailed in Sect 3.3. So we would like to maintain that our statements are not speculative, but based on comprehensible arguments.

**Fig. 5 and ll. 373 ff.:** How were the time series reconstructed (dashed lines)? I understand that peak water level and hence peak flow can be reconstructed from debris lines etc., but how about the recession? How can we explain the high water level at gauge Schönau after the peak, while discharge had already recessed to normal levels?

On the general reconstruction of the waterlevel and discharge time series, please see our reply to RC Ll. 153 ff.:. In particular about the recession limb: Sometimes, water level recordings were not available during the flood, but for a few points in time, water level reconstruction was possible from photos available after the flood, plus temporal interpolation. In other cases, recession limbs could be estimated from recession limbs from upstream gauges with observations. Waterlevel recordings or estimates could then be used to estimate discharge. For this, often pre-event W-Q relations could not be used because of substantial changes of the river cross-section (erosion or deposition). This is for example the case for gauge Schönau, where deposition lead to continuously high water levels after the flood despite declining discharge. Such effects were detected by the responsible water authorities and considered in the reconstruction of discharge. Now, we addressed this in the (new) data section.

**Ll. 456 ff. / section 3.3:** many statements in section 3.3 are not identifiably based on scientific data - either observation or models -, but seem to be speculative, hear-say, or partly reproducing textbook knowledge. The actual scientific contribution of this section remains unclear to me, especially in comparison to the paper of Dietze et al. (2022) which is widely cited by the present study.

It is nearly impossible to obtain meaningful quantitative data (not casuistic but enough data and through a minimum sampling scale), and to go for a modelling approach would take an effort not compatible with this manuscript. With the analysis of the sparse data which is available in terms of morphology changes in the valley, we aimed at (i) clearly demonstrate the importance of the hydro-morphodynamic processes in the enhancement of the valley hazard; (ii) to clearly show how the anthropic modification of the catchment enhances the flood hazard; (iii) to identify and explain how morphology (natural or anthropic) singularities of the valley become focus of hazard during floods; (iv) how hydrodynamic and geomorphic processes explain the damages which we observed in a post event analysis. We understand that the objectives of the paper by our colleagues Dietze et al. (2022) has somehow commonalities, however we (i) go further in aspects related to the understanding of the anthropic modification of the valley (mainly of urban and industrial character), (ii) we present complementary and novel information and cases and (iii) our interpretation fills some gaps or questions the interpretations by our colleagues.

We edited the section significantly in the new version of the paper (and also shortened it to make our own scientific contribution clearer). In addition, we added a few introductory sentences to make our approach/objectives (or the problem with the data) clearer.

**Fig. 9:** I think that the scale of this figure is not really helpful for appreciating the inundated area.
Figure 9 (now 10) was revised. Especially the cities/village names should now help for orientation. Please keep in mind that this is a general map and that the Ahr is only 2 to 4 m wide.

**Ll. 589 / section 4.1:** I think the issue of rapid inundation mapping is an important research topic, but I do not see how it is relevant in the post-event analysis almost one year after the fact.
See answers above (new research questions inclining new section (Synopsis)). Furthermore, we modified the section.

**Ll. 630 ff. / section 4.2:** The same applies to section 4.2: Why is the rapid damage estimation process important? In the context of this study, shouldn't the best possible estimates of damages and damage processes be used?
See answers above (new research questions inclining new section (Synopsis)). Furthermore, we modified the section. Furthermore, section 3.3 emphasised very much the role of debris/sediment/hydro-morphodynamics for damage processes, but I do not find this issue in section 4.2. Isn't this damage process chain one of the key properties of the July 2021 event?
See our answers above (new section 5 and new research questions)

**Ll. 700 ff. / section 4.3:** This section is mainly based on the number of traffic reports and the number of affected railroad sections, irrespective of the severity of disruption. Doesn't that limit the meaningfulness and the comparability to the 2013 event, given that it should matter whether a road/bridge is just disrupted or whether it actually disappeared? Maybe the persistence of the disruption for specific sections/lines could be interpreted as a proxy for severity, but there should be a reference from other events to compare to.
It would certainly be interesting to examine this in more detail; but unfortunately we do not have any information on the severity, so a comparison is difficult; we can only make statements about this event and are limited in the details.

**Fig. 11b:** In total, I find subplot b not very informative, particularly since most of the specific road and railway numbers are not referenced in the main text. Wouldn't a simple plot of affected sections/lines over time be more informative and concise?
In line with your comments at the beginning, we shorten this section and moved the figure to the supplement, as the main messages are understandable without the figure.

**Ll. 744 ff:** *"The July 2021 flood in western Germany and neighbouring regions was one of the five most severe and expensive natural catastrophes in Europe in the last half century"* - what is the source of that statement?
This is our result (Based on our Database CATDAT; see section 4.2. line 696 now Sect. 2.7 line 250)). We rewrote this.

How important is the consideration of EFI in the context of your study, given the apparently high skill of the ICON models?
The point here is that not only the ICON but also other models (here: ECMWF-ENS) predict the extreme event. This again speaks for the general good predictability of the event (from the meteorological side) and that not only one model had this event on the screen.

**L. 763:** *"However, because of the high political relevance, the failure of the warning chain was not discussed here."* I understand, but why bring it up then? Could in this case the topic of predictability also be dropped from the manuscript?
As this is an important part in the context of the synopsis section and the new timeline (sequence of events), we did not delete this assessment. Nevertheless, this passage was deleted in the new version (due to the reduction of the text).

**L. 809:** *"Remote sensing of rapidly available imagery from social media, television, and news media [...]"* - I don't understand what that means.
This sentence (resp. the passage) is no longer part of the paper.

**Ll. 817 ff.:** *"[...] our still insufficient knowledge must be further improved through more dedicated research"* - is it really necessary to spell out the obvious?
Yes, it cannot be emphasized often enough. We rewrite the sentence.

**Ll. 823 ff.:** Is this item a new conclusion drawn from this study? Wasn't this already pointed out by Roggenkamp and Herget (2022)?
Based on our work, we both came to the same conclusion (cf. Schäfer et al., 2021; KIT, 2021); ultimately, this point cannot be emphasized often enough; moreover, the last point is also very important for us to emphasize. At this point, we have once again explicitly referred to Roggenkamp.

**Ll. 849 ff.:** Sure, but how??
We added a sentence: The use of video analysis can be an alternative, allowing distance observation of flood levels and inference of flood discharges with fairly good results (Detert et al., 2017).

**Ll. 853 ff:** How is this a conclusion from the present study?
In line with our new RQ2, we revised this finding.

**Ll. 876 ff.:** Please see my above comment: I think it would be really helpful to publish the data along with this manuscript.
See answer above.

**Technical comments**

Thank you for your carefully reading; unless otherwise indicated, we will consider all the comments below.

**L. 160:** Please replace "flood plain" by "inundated areas"
Thanks, we will change this to "inundation areas" in consistence with the whole text.

**L. 189:** were used

**L. 191:** 95th percentile

**L. 210:** suitability instead usability

**L. 224:** second "decade"?
We reformulated this: "On 10\,}July 2021, a prominent upper-level trough…"

**Ll. 272 ff.:** *"The long-term average for the month of July at this station is 69 mm (1981-2010); thus, in just a few hours, the rainfall added up to more than twice the usual monthly precipitation."* - This kind of statement is unnecessary.
We will delete the second part.

**L. 307:** as early as

**L. 355:** replace same by similar

**L. 580:** replace "the hardest affected location" by "the location affected most severely"

**Fig. 10:** It impossible to distinguish EMSR517_v1 from EMSR517_v3
We revised the figure.

**Fig. 11:** the resolution is very low.
Due to the comments above, we have decided to no longer use the figure in the main document (see above).

**L. 762:** delete "the occurrence"

**L. 831:** an extraordinary

**L. 837:** the term "re-forecast" sounds weird - isn't "hindcast" the correct term?
Both is common; for example, re-forecast is commonly used by the ECMWF:
https://www.ecmwf.int/en/forecasts/documentation-and-support/extended-range/re-forecast-medium-and-extended-forecast-range
https://www.ecmwf.int/en/forecasts/datasets/set-vi

**Reviewer #2**

**General comments**

We thank the reviewer for his/her time and comments, corrections, and suggestions.

The paper "A multi-disciplinary analysis of the exceptional flood event of July 2021 in central Europe. Part 1: Event description and analysis" by Mohr et al. gives an overview of the flood event last year, with a special focus on hydrological, and hydro-morphological processes and mechanisms. The paper describes very well the event across various physical disciplines. The complex interaction between those could be analyzed in more detail.
We introduced a new section called "Synopsis" to better handle this point (including a new figure). More details can also be found in our responses to Reviewer 1, who also addressed this point.

The aspects of social science regarding the flood event are not addressed, at least the paper should underline of refer to the high importance of risk culture (e.g. risk awareness, risk communication).
This is difficult to take into account; first, the other reviewer points out that our work should be more stringent and we should concentrate more on our own results with clearly defined questions (and not be too broad in the text). Second, this aspect (risk culture) is not our objective of the study, as we did not do own research on this topic. However, we briefly addressed this aspect in the conclusion and referred to appropriate work by others.

**Specific comments**

L 1; L 16; L 69; L334, L744: Please think about if you want to use the term "natural disaster". There is no disaster without human interference, so it's never something "natural" Have a look at #nonaturaldisaster: https://www.nonaturaldisasters.com/
That is of course correct; we will adapt that (either "natural hazard" or only "disaster").

L 23: Figure 1 is mentioned here for the first time, but Figure 1 is currently in L 116. Why so far away?
We can move the figure to the introduction, but in the end the publisher's typesetters will decide the position.

L 35: displaced people? Yes, we will add "people".

L 39: in the meantime, flood hazard maps are updated see Roggenkamp & Herget. I would rather write the existing maps before and during the flood
Yes, that's right, we know that these have been updated in the meantime; therefore we will use the suggested formulation.

L 107: see below – could you describe below more specific please?
In fact this reference is irrelevant (or unnecessary), we will delete it.

**Technical corrections**

Thank you for the carefully reading; unless otherwise indicated (e.g., numbers), we will consider these technical corrections.

L105: one hour, but can reach up to one minute → Numbers from one to twelve are written out
According to the NHESS guidelines (https://www.natural-hazards-and-earth-system-sciences.net/submission.html), this would in general be correct (..."use words for cardinal numbers less than 10"...); however, for items that are "units of time or measure", this is not the case, and here this is a time unit.

L 38: only one week
See above

L63: six month
See above

L73; L74: (e.g. ....)

L74: erosion, and

L128: used by

L129: In its global uniform resolution configuration it is run twice daily → check the grammar
We will include a comma

L307: as early as

L310: were predicted more than two days
See above

L320: two days
See above

L339: - namely soil wetness → missing spaces

L344: three weeks
See above

L371: erosion, and

L372; L389f; L401, L484: - …. → missing spaces

L408f: the peak flow

L523: floods are?

L632: infrastructural, and

L762: two days
See above

L857: This helps to mitigate associated adverse effects -> missing . in the end